# Understanding and Mitigating Memorization in Diffusion Models for Tabular Data

**Zhengyu Fang** [* 1] **Zhimeng Jiang** [* 2] **Huiyuan Chen** [1] **Xiao Li** [1 3 4 5] **Jing Li** [1]

## Abstract

Tabular data generation has attracted significant research interest in recent years, with the tabular diffusion models greatly improving the quality of synthetic data. However, while memorization—where models inadvertently replicate exact or near-identical training data—has been thoroughly investigated in image and text generation, its effects on tabular data remain largely unexplored. In this paper, we conduct the first comprehensive investigation of memorization phenomena in diffusion models for tabular data. Our empirical analysis reveals that memorization appears in tabular diffusion models and increases with larger training epochs. We further examine the influence of factors such as dataset sizes, feature dimensions, and different diffusion models on memorization. Additionally, we provide a theoretical explanation for why memorization occurs in tabular diffusion models. To address this issue, we propose TabCutMix, a simple yet effective data augmentation technique that exchanges randomly selected feature segments between random same-class training sample pairs. Building upon this, we introduce TabCutMixPlus, an enhanced method that clusters features based on feature correlations and ensures that features within the same cluster are exchanged together during augmentation. This clustering mechanism mitigates out-of-distribution (OOD) generation issues by maintaining feature coherence. Experimental results across various datasets and diffusion models

demonstrate that TabCutMix effectively mitigates memorization while maintaining high-quality data generation. Our code is available at `https://github.com/fangzy96/TabCutMix`.

## 1. Introduction

Tabular data generation has gained increasing attention due to its broad applications, such as data imputation (Zheng & Charoenphakdee, 2022; Liu et al., 2024; Villaizán-Vallelado et al., 2024), data augmentation (Fonseca & Bacao, 2023), and data privacy protection (Zhu et al., 2024; Assefa et al., 2020). Unlike image or text data, tabular data consists of structured datasets commonly found in fields such as healthcare (Hernandez et al., 2022), finance (Assefa et al., 2020), and e-commerce (Cheng et al., 2023). Its heterogeneous and mixed-type feature space often poses unique challenges for generative models (Yang et al., 2024b; Zhang et al., 2023b). Recent advances have led to the development of various methods aimed at improving the quality of synthetic tabular data, with diffusion models emerging as a particularly effective approach (Zhang et al., 2023a; Kotelnikov et al., 2023). These models have demonstrated significant improvements in generating high-quality tabular data, making them a powerful tool for a wide range of applications.

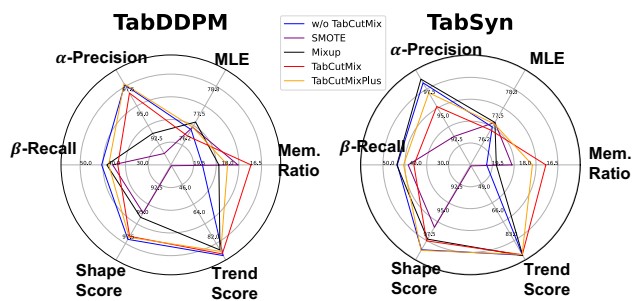

*Figure 1.* The overview performance of TabCutMix in TabDDPM and TabSyn for Default dataset. "Mem. Ratio" represents the memorization ratio.

Despite these advancements, an often-overlooked issue is the phenomenon of memorization, where diffusion models unintentionally replicate exact or nearly identical samples from the training data. This not only introduces pri-

---

[*]Equal contribution [1]Department of Computer and Data Sciences, Case Western Reserve University, Cleveland, USA [2]Department of Computer Science & Engineering, Texas A&M University, College Station, USA [3]Department of Biochemistry, Case Western Reserve University, Cleveland, USA [4]Center for RNA Science and Therapeutics, Case Western Reserve University, Cleveland, USA [5]Department of Biomedical Engineering, Case Western Reserve University, Cleveland, USA. Correspondence to: Jing Li <jingli@cwru.edu>.

*Proceedings of the 42nd International Conference on Machine Learning*, Vancouver, Canada. PMLR 267, 2025. Copyright 2025 by the author(s).

vacy concerns but also hampers model generalization (Yoon et al., 2023; Kandpal et al., 2022). While this phenomenon has been extensively investigated in image and text generation (Karras et al., 2022; Carlini et al., 2021; Song et al., 2021; Ho et al., 2020), its occurrence and impact in tabular data generation remain relatively unexplored. This gap in understanding leads to a key question:

**Does memorization occur in tabular diffusion models, and if so, how can it be effectively mitigated?**

In this paper, we aim to address this gap by conducting the first comprehensive investigation into memorization behaviors within tabular diffusion models. Through rigorous empirical analysis, we examine how various factors—such as training dataset sizes, feature dimensions, and model architecture—affect the extent of memorization. Additionally, we provide a theoretical exploration of memorization in tabular diffusion models, shedding light on the underlying mechanisms that lead to the issue of memorization in tabular data.

To mitigate memorization, we firstly introduce **TabCutMix**, a simple yet effective data augmentation technique that swaps randomly selected feature segments between training samples within the same class. Building on this, we propose **TabCutMixPlus**, an enhanced augmentation method that clusters features based on feature correlations and ensures that features within the same cluster are exchanged together. This clustering mechanism not only mitigates memorization but also mitigates OOD generation challenges by maintaining feature coherence during augmentation. Extensive experiments across multiple datasets and diffusion models demonstrate that TabCutMixPlus outperforms TabCutMix and other baseline methods in reducing memorization (See Figure. 1) without compromising the quality of the synthetic data, making it a practical solution for improving tabular data generation in real-world scenarios.

## 2. Related Work

**Tabular Generative Models.** Generative models for tabular data have gained attention due to their broad applicability. Early approaches like CTGAN and TVAE (Xu et al., 2019) leveraged Generative Adversarial Networks (GANs) (Goodfellow et al., 2020) and VAEs (Kingma, 2013) for handling imbalanced features. GOGGLE (Liu et al., 2023) advanced this by modeling feature dependencies using graph neural networks. Inspired by NLP advancements, GReaT (Borisov et al., 2023) transformed rows into natural language sequences to capture table-level distributions. More recently, diffusion models, originally successful in image generation (Ho et al., 2020), have been adapted for tabular data, as demonstrated by STaSy (Kim et al., 2023), TabDDPM (Kotelnikov et al., 2023), CoDi (Lee et al., 2023),

TabSyn (Zhang et al., 2023a), and balanced tabular diffusion (Yang et al., 2024b).

**Memorization in Generative Models.** Memorization has been widely studied in image and language domains (van den Burg & Williams, 2021; Gu et al., 2023; Huang et al., 2024). In image generation, reseasrchers (Somepalli et al., 2023a; Carlini et al., 2021) found that diffusion models, like Stable Diffusion (Rombach et al., 2022) and DDPM (Ho et al., 2020), memorize portions of their training data at varying levels. Concept ablation (Kumari et al., 2023) is proposed to mitigate memorization via fine-tuning of pre-trained models to minimize output disparity. AMG (Chen et al., 2024) uses real-time similarity metrics to selectively apply guidance to likely duplicates. For text generation, text conditioning amplifies memorization risks, especially in large-scale language models (Somepalli et al., 2023a;b; Huang et al., 2024). Goldfish loss (Hans et al., 2024) randomly drops a subset of tokens from the training loss computation to prevent the model from memorizing. Memorization prediction (Biderman et al., 2024), i.e., predicting which sequences will be memorized before full-scale training, is investigated by analyzing the memorization patterns of lower-compute trial runs for early intervention. Although these patterns are evident in image and text generation, the impact of memorization on tabular data remains underexplored.

## 3. Memorization in Tabular Diffusion Models

Despite the development of numerous high-performing diffusion models for tabular data generation, it remains unclear whether these models are susceptible to memorization. In this section, we introduce a criterion for detecting and quantifying the intensity of memorization in tabular data. Using this criterion, we explore memorization behaviors across various diffusion models under different dataset sizes and feature dimensions. We choose two state-of-the-art (SOTA) generative models: TabSyn (Zhang et al., 2023a) and TabDDPM (Kotelnikov et al., 2023) for our preliminary memorization analysis. Furthermore four real-world tabular datasets—Adult, Default, Shoppers, and Magic—each containing both numerical and categorical features are included. The details of the datasets can be found in Section 5. Additionally, we provide a theoretical analysis to explain the mechanisms behind memorization in tabular diffusion models.

### 3.1. Memorization Detection Criterion

A quantitative criterion is essential for quantifying the memorization ratio—i.e., the proportion of generated samples that are memorized by a model. In natural language processing, memorization is typically identified when a model

can reproduce *verbatim* sequences from the training set in response to an adversarial prompt (Carlini et al., 2021; Kandpal et al., 2022). However, such a verbatim definition is not directly applicable to image and tabular data, where the intrinsic continuous nature of pixels and features makes exact replication less meaningful.

Inspired by prior work in image generation (Yoon et al., 2023; Gu et al., 2023), we adopt the "relative distance ratio" criterion to detect whether a generated sample $x$ is a memorized replica from training data $\mathcal{D}$ in tabular dataset. Specifically, $x$ is considered memorized if $d\big(x, \mathrm{NN}_1(x, \mathcal{D})\big) < \frac{1}{3} \cdot d\big(x, \mathrm{NN}_2(x, \mathcal{D})\big)$, where $d(\cdot, \cdot)$ is the distance metric in the input sample space, $\mathrm{NN}_i(x, \mathcal{D})$ represents $i$-th nearest neighbor of $x$ in training data $\mathcal{D}$ based on the distance $d(\cdot, \cdot)$[1].

In the image generation domain, $l_2$ norm is commonly adopted as the distance metric to measure the sample similarity in the input space. However, this metric is not suitable for tabular data generation due to the mix-typed (categorical and numerical) input features. To address this, and inspired from mixed-type data clustering literature (Ji et al., 2013; Ahmad & Khan, 2019), we define a mixed distance $d(\cdot, \cdot)$ between generated sample $x$ and real training sample $x'$ as follows:

$$d(x, x') = \frac{1}{M} \left( \mathrm{norm}\left( \sqrt{\sum_{i \in \mathcal{F}_{num}} (x_i - x'_i)^2} \right) + \sum_{j \in \mathcal{F}_{cat}} \mathbf{1}(x_j \neq x'_j) \right). \quad (1)$$

where $\mathcal{F}_{num}$ and $\mathcal{F}_{cat}$ represent the index sets for numerical and categorical features, respectively; $\mathrm{norm}(d_n)$ represents max-min normalization rescaling the distance values to a $[0, 1]$ range using $\mathrm{norm}(d_k) = \frac{d_k - \min_k(d_k)}{\max_k(d_k) - \min_k(d_k)}$, where $k$ is sample pair distance index; $M$ is the total number of features, such that $|\mathcal{F}_{num}| + |\mathcal{F}_{cat}| = M$. In this equation, $x_i(x'_i)$ represents $i$-th feature value for sample $x(x')$, $\mathbf{1}(x_j \neq x'_j)$ is an indicator function that equals 1 if $x_j \neq x'_j$ and 0 otherwise. In this paper, we use Eq. (1) to measure sample similarity and to quantify the memorization ratio in tabular data generation.

## 3.2. Effect of Different Diffusion Models

In this subsection, we focus on examining the behavior of the two diffusion models (TabSyn (Zhang et al., 2023a) and TabDDPM (Kotelnikov et al., 2023)) on the memorization ratio across the four tabular datasets (Adult, Default,

---

[1]The factor $\frac{1}{3}$ is an empirical threshold and widely adopted in image generation literature. Mem-AUC is also defined in Appendix D.6

Shoppers, and Magic). For each dataset, we check the memorization ratio over the course of training of TabSyn and TabDDPM. Figure 2 illustrates the memorization ratio for both models. Based on our experiments, we make the following observations:

**Obs.1:** TabSyn exhibits faster convergence with more stable memorization ratios across all datasets compared to TabD-DPM. This trend is particularly prominent for the Default and Adult datasets, where TabSyn stabilizes its memorization rate after approximately 500 epochs, while TabDDPM continues to fluctuate over a much longer training duration, up to 4000 epochs.

**Obs.2:** Although the converged memorization rates vary between datasets, the final memorization levels are relatively similar across both diffusion models. For instance, in TabSyn, the memorization ratio for Magic can reach up to 80%, indicating high memorization, whereas it stabilizes at 20% in Default, showing lower memorization. Similar trends are observed in TabDDPM, suggesting that while the training dynamics differ, the overall memorization capacity converges to comparable levels across models for the same dataset.

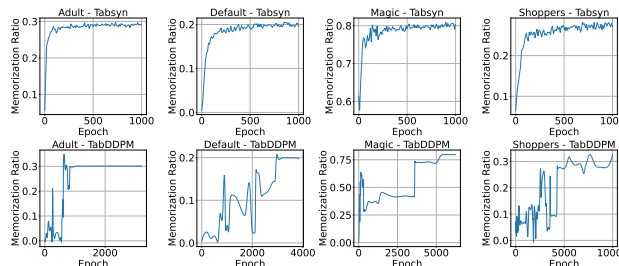

*Figure 2.* Memorization ratio curve of TabSyn and TabDDPM w.r.t. training epochs.

## 3.3. Impact of Training Dataset Size

Building on the findings from Section 3.2, where TabSyn demonstrated high training stability, we use TabSyn as the backbone model to explore the impact of training dataset size on memorization in tabular data. We conduct experiment with four datasets (Default, Shoppers, Magic, and Adult), randomly downsampling the training samples to five different sizes: 0.1%, 1%, 10%, 50%, and 100% of the original dataset. Figure 3 shows the memorization ratio for each dataset size over the training epochs. We make the following observations:

**Obs.1**: Smaller training datasets consistently exhibit higher memorization ratios, as observed across all datasets when the training size is reduced to 0.1%. For some datasets, such as Shoppers, even moderate reductions in training size (e.g., 10%) lead to noticeable increases in memorization, whereas for others, such as Magic, the effect becomes prominent

only at extremely small sizes (e.g., 0.1%).

**Obs.2**: The memorization ratio generally increases over training epochs before stabilizing. The final converged memorization ratio demonstrates a strong dependency on training dataset size when the size is extremely small (e.g., 0.1%). For larger sizes, such as 10%, the dependency is less pronounced for datasets like Magic and Shoppers, possibly due to the relatively larger sample pool. This observation suggests that the impact of dataset size on memorization becomes increasingly critical as the dataset size decreases.

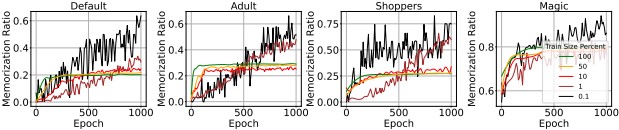

*Figure 3.* Impact of dataset size among different datasets for Tab-Syn model.

### 3.4. Theoretical Analysis

In the previous section, we empirically investigate the memorization phenomenon in existing tabular diffusion models. However, the underlying cause of memorization in tabular diffusion models remains unclear. To bridge the gap, we applied theoretical analysis from image generation (Gu et al., 2023) to rationalize why memorization occurs in TabSyn (Zhang et al., 2023a), one of the SOTA tabular generative models.

In TabSyn, a variational autoencoder (VAE) is used to map the input features $\boldsymbol{x}$ into an embedding $\boldsymbol{z} = \text{Encoder}(\boldsymbol{x})$ in latent space. Subsequently, a latent diffusion is applied to generate samples in the latent space. The final synthetic data is generated via the decoder of VAE. For simplicity, we only consider latent diffusion in the analysis. Specifically, the following forward and backward stochastic differential equations are adopted in the latent diffusion:

$$
\begin{aligned}
\boldsymbol{z}_t &= \boldsymbol{z}_0 + \sigma(t)\boldsymbol{\epsilon}, \boldsymbol{\epsilon} \sim \mathcal{N}(\mathbf{0}, \boldsymbol{I}), \quad (2) \\
\mathrm{d}\boldsymbol{z}_t &= -2\dot{\sigma}(t)\sigma(t)\boldsymbol{s}(\boldsymbol{z}_t, t)\mathrm{d}t + \sqrt{2\dot{\sigma}(t)\sigma(t)}\mathrm{d}\boldsymbol{\omega}_t, (3)
\end{aligned}
$$

where $\boldsymbol{z}_0 = \boldsymbol{z}$ represents the initial embedding from the encoder, $\boldsymbol{z}_t$ is the diffused embedding at time $t$, and $\sigma(t)$ is the noise level at time $t$. The score function $\boldsymbol{s}(\boldsymbol{z}_t, t)$ is defined as $\boldsymbol{s}(\boldsymbol{z}_t, t) = \nabla_{\boldsymbol{z}_t} \log p_t(\boldsymbol{z}_t)$, and $\boldsymbol{\omega}_t$ is the standard Wiener process.

When the score function $\boldsymbol{s}(\boldsymbol{z}_t, t)$ is known, synthetic data can be sampled by reversing the diffusion process. In practice, diffusion models train a neural network $\boldsymbol{s}_\theta(\boldsymbol{z}_t, t)$ to approximate the score function $\boldsymbol{s}(\boldsymbol{z}_t, t)$. However, score function $\nabla_{\boldsymbol{z}} \log p_t(\boldsymbol{z})$ is intractable since the marginal distribution $p_t(\boldsymbol{z}) = p(\boldsymbol{z}_t)$ is unknown. Fortunately, the conditional distribution $p(\boldsymbol{z}_t|\boldsymbol{z}_0)$ is tractable and can be used to train the denoising function to approximate the conditional score

function $\nabla_{\boldsymbol{z}_t} \log p(\boldsymbol{z}_t|\boldsymbol{z}_0)$. The denoising score-matching training process is formulated as:

$$
\min \mathbb{E}_{\boldsymbol{z}_0 \sim p(\boldsymbol{z}_0)} \mathbb{E}_{\boldsymbol{z}_t \sim p(\boldsymbol{z}_t|\boldsymbol{z}_0)} \left\| \boldsymbol{s}_\theta(\boldsymbol{z}_t, t) - \nabla_{\boldsymbol{z}_t} \log p(\boldsymbol{z}_t|\boldsymbol{z}_0) \right\|_2^2.
$$
$$(4)$$

where $\nabla_{\boldsymbol{z}_t} \log p(\boldsymbol{z}_t|\boldsymbol{z}_0)$ can be calculated according to $\nabla_{\boldsymbol{z}_t} \log p(\boldsymbol{z}_t|\boldsymbol{z}_0) = -\frac{\boldsymbol{\epsilon}}{\sigma(t)}$.

Regarding memorization of synthetic data in the latent space, we have

**Proposition 3.1** ( (Gu et al., 2023)). *Assume that the neural network can perfectly approximate the optimal score function $\boldsymbol{s}_\theta^*(\boldsymbol{z}_t, t)$ given by Eq. (6) and a perfect SDE solver is applied in backward SDE. The generated sample in latent space $\boldsymbol{z}_0$ will exactly replicate the latent embedding of the real sample in training data.*

See proof in Appendix. B. Proposition. 3.1 demonstrates that under ideal conditions, the generated sample in latent space is an exact representation of a real training sample, which contradicts the empirical observation in TabSyn (i.e., not 100% memorization). There are several possible reasons for this discrepancy. First, the practical score-matching function learned by the neural network may not perfectly approximate the optimal score due to insufficient optimization or limited model capacity. Additionally, TabSyn uses a VAE to handle mixed-type tabular data, followed by latent diffusion for generation. As a result, even if the generated sample in latent space is identical to a training sample, the final generated sample may differ due to the randomness introduced by the VAE decoder.

## 4. Methodology

Building on the memorization study presented in Section 3, we identify that the memorization in tabular diffusion models remains a significant yet underexplored issue, limiting the diversity and utility of generated data. To address this problem, we propose two novel data augmentation strategies tailored for tabular data generation [2]: TabCutMix and its enhanced version, TabCutMixPlus. The pseudo-code of our proposed algorithm is in Appendix C.

### 4.1. TabCutMix

TabCutMix generates a new training sample $(\tilde{\boldsymbol{x}}, \tilde{y})$ by combining two samples $(\boldsymbol{x}_A, y_A)$ and $(\boldsymbol{x}_B, y_B)$ that belong to the *same* class. In tabular data, "same class" refers to instances that share the same categorical target label, ensuring that the generated sample remains consistent with its original class and prevents label mismatches. The newly gener-

---

[2]These strategies are inspired by the CutMix (Yun et al., 2019) data augmentation technique used in the image domain

ated sample is defined using the following mix operation:

$$\tilde{\boldsymbol{x}} = \boldsymbol{M} \odot \boldsymbol{x}_A + (\boldsymbol{1} - \boldsymbol{M}) \odot \boldsymbol{x}_B, \qquad (5)$$

where $\boldsymbol{M} \in \{0, 1\}^M$ is a binary mask matrix indicating which features to swap between the two samples, $\boldsymbol{1}$ is a mask filled with ones, and $\odot$ represents element-wise multiplication. The portion of exchanged features $\lambda$ is sampled from the uniform distribution $\mathcal{U}(0, 1)$. Each element of $\boldsymbol{M}$ is sampled independently from a Bernoulli distribution $\text{Bern}(\lambda)$. In each training iteration, we first sample the class index $c \in \{1, 2, \cdots, C\}$ using the class prior distribution and then randomly select two samples from that class.

### 4.2. TabCutMixPlus

While TabCutMix significantly reduces memorization, it may inadvertently disrupt inter-feature relationships, particularly in highly correlated features. Such disruptions can lead to the generation of out-of-distribution (OOD) samples, thereby reducing the reliability of the synthetic data. To overcome this limitation, we propose TabCutMixPlus, an advanced augmentation strategy that preserves structural integrity by clustering features based on their correlations and performing swaps within clusters.

TabCutMixPlus identifies clusters of highly correlated features using domain-specific correlation measures and hierarchical clustering algorithm[3]. For numerical features, we use the Pearson correlation coefficient, while for categorical features, we employ Cramér's V (Cramér, 1999). For numerical-categorical feature pairs, we calculate the squared ETA coefficient (Richardson, 2011). Each cluster is treated as an atomic unit during the augmentation process, ensuring that only features within the same cluster are exchanged together. This clustering approach maintains the relationships among highly correlated features, thereby mitigating the risk of generating OOD samples.

By ensuring feature coherence during augmentation, Tab-CutMixPlus strikes a balance between reducing memorization and maintaining high-quality synthetic data. Extensive experiments on OOD detection, detailed in Appendix D.4.6, demonstrate that TabCutMixPlus significantly outperforms TabCutMix and generates superior data utility.

## 5. Experiments

In this section, we extensively evaluate the effectiveness of TabCutMix and TabCutMixPlus across several SOTA tabular diffusion models in various datasets and compared other augmentation methods Mixup (Zhang, 2017; Takase, 2023) and SMOTE (Chawla et al., 2002).

### 5.1. Experimental Setup

**Datasets.** We use four real-world tabular datasets containing both numerical and categorical features: Adult Default, Shoppers, and Magic. The detailed descriptions and overall statistics of these datasets are provided in Appendix D.1.

**Diffusion Models.** We integrate TabCutMix with three existing SOTA diffusion-based tabular data generative models, including TabDDPM (Kotelnikov et al., 2023), STaSy (Kim et al., 2023), and TabSyn (Zhang et al., 2023a). To the best of our knowledge, this work is the first to comprehensively evaluate both generation quality and memorization performance for these models.

**Evaluation Metrics.** We evaluate the performance of synthetic data generation from two perspectives: memorization and synthetic data quality. For memorization evaluation, we generate the same number of synthetic samples as the training dataset and use Eq. (1) to calculate the distance between the generated and real samples. The generated sample is considered memorized if its closest neighbor in the training data is less than $\frac{1}{3}$ of the distance to its second closest neighbor (Yoon et al., 2023; Gu et al., 2023). Memorization ratio is defined as the proportion of generative samples that are memorized, using a fixed threshold of $\frac{1}{3}$. To complement this, we introduce *Mem-AUC* in Appendix D.6, which summarizes memorization behavior by averaging the memorization ratio across a continuous range of thresholds. This metric provides a more comprehensive and robust evaluation, especially when the memorization behavior may vary under different threshold settings. To validate the use of the fixed $\frac{1}{3}$ threshold in practice, we compute both Mem-AUC and the memorization ratio at $\frac{1}{3}$, and analyze their correlation in Appendix E.9. The results reveal a strong positive correlation, indicating that the fixed-threshold metric serves as a reliable proxy for the more holistic Mem-AUC. Furthermore, as shown in Figure 11, we also compute the correlations among memorization ratios under thresholds $\frac{1}{2}$, $\frac{1}{3}$, and $\frac{1}{4}$, and observe consistently high correlations across all threshold pairs. This further supports the robustness of the memorization metric under different threshold choices, and highlights $\frac{1}{3}$ as a representative and stable threshold that balances simplicity and practical effectiveness. For synthetic data quality evaluation, we consider 1) low-order statistics (i.e., column-wise density and pair-wise column correlation) measured by shape score[4] and trend score [5]; 2) high-order metrics $\alpha$-precision and $\beta$-recall scores measuring the over-

---

[3]https://docs.scipy.org/doc/scipy/reference/cluster.hierarchy.html

[4]Shape Score measures how closely the synthetic data matches the distribution of individual columns in the real data using Kolmogorov-Smirnov (KS) test.

[5]Trend Score assesses whether the relationships or correlations between pairs of columns in the synthetic data are similar to those in the real data

*Table 1.* The overview performance comparison for tabular diffusion models on more datasets. "TCM" represents our proposed **TabCut-Mix** and "TCMP" represents **TabCutMixPlus**. "Mem. Ratio" represents memorization ratio. "Improv" represents the improvement ratio on memorization.

| | Methods | Mem. Ratio (%) ↓ | **Improv.** | MLE (%)↑ | α-Precision(%)↑ | β-Recall(%)↑ | Shape Score(%)↑ | Trend Score(%)↑ | C2ST(%)↑ | DCR(%) |
|---|---|---|---|---|---|---|---|---|---|---|
| **Default** | STaSy | 17.57 ± 0.53 | - | 76.48 ± 1.18 | 87.78 ± 5.20 | 35.94 ± 5.48 | 90.27 ± 2.43 | 89.58 ± 1.35 | 67.68 ± 6.89 | 50.30 ± 0.36 |
| | STaSy+Mixup | 17.89 ± 0.99 | −1.80% ↓ | 75.69 ± 1.26 | 82.65 ± 10.01 | 37.94 ± 2.57 | 85.77 ± 4.02 | 86.49 ± 4.66 | 50.81 ± 6.01 | 50.66 ± 1.39 |
| | STaSy+SMOTE | 15.98 ± 0.04 | 9.07% ↓ | 75.41 ± 0.95 | 86.75 ± 5.80 | 32.95 ± 2.93 | 87.89 ± 5.17 | 32.54 ± 0.91 | 48.57 ± 5.90 | 51.39 ± 2.23 |
| | STaSy+TCM | 14.51 ± 0.46 | 17.44% ↓ | 75.33 ± 1.32 | 86.04 ± 11.55 | 32.13 ± 5.07 | 90.30 ± 3.88 | 89.85 ± 3.16 | 49.51 ± 6.33 | 50.39 ± 0.99 |
| | STaSy+TCMP | 15.53 ± 2.00 | 11.59% ↓ | 76.30 ± 0.57 | 90.83 ± 4.51 | 32.81 ± 1.37 | 91.49 ± 0.77 | 92.08 ± 2.04 | 50.43 ± 2.00 | 50.70 ± 1.94 |
| | TabDDPM | 19.33 ± 0.45 | - | 76.79 ± 0.69 | 98.15 ± 1.45 | 44.41 ± 0.70 | 97.58± 0.95 | 94.46 ± 0.68 | 91.85 ± 6.04 | 49.12 ± 0.94 |
| | TabDDPM+Mixup | 18.46 ± 0.71 | 4.50% ↓ | 77.18 ± 0.35 | 93.20 ± 4.16 | 42.59 ± 1.13 | 95.34 ± 1.79 | 90.32 ± 3.31 | 92.59 ± 2.82 | 52.36 ± 1.57 |
| | TabDDPM+SMOTE | 17.46 ± 0.51 | 9.66% ↓ | 76.92 ± 0.35 | 91.19 ± 0.68 | 40.52 ± 0.65 | 94.89 ± 1.46 | 28.63 ± 2.28 | 72.73 ± 0.69 | 50.95 ± 0.38 |
| | TabDDPM+TCM | 16.76 ± 0.47 | 13.26% ↓ | 76.47 ± 0.60 | 97.30 ± 0.46 | 38.72 ± 2.78 | 97.27 ± 1.74 | 93.27 ± 2.52 | 94.72 ± 3.87 | 50.23 ± 0.53 |
| | TabDDPM+TCMP | 18.00 ± 0.24 | 6.88% ↓ | 76.92 ± 0.17 | 98.26 ± 0.25 | 41.92 ± 0.52 | 97.37 ± 0.09 | 91.42 ± 1.15 | 95.64 ± 0.49 | 49.75 ± 0.32 |
| | TabSyn | 20.11 ± 0.03 | - | 77.00 ± 0.33 | 98.66 ± 0.13 | 46.76 ± 0.50 | 98.96 ± 0.11 | 96.82 ± 1.71 | 98.27 ± 1.14 | 51.09 ± 0.32 |
| | TabSyn+Mixup | 19.58 ± 0.33 | 2.65% ↓ | 77.24 ± 0.42 | 99.05 ± 0.45 | 46.94 ± 0.19 | 97.84 ± 0.16 | 97.11 ± 0.42 | 96.82 ± 1.99 | 49.80 ± 0.17 |
| | TabSyn+SMOTE | 18.72 ± 0.54 | 6.93% ↓ | 77.24 ± 0.43 | 93.00 ± 0.29 | 42.78 ± 0.64 | 96.59 ± 0.10 | 32.70 ± 0.23 | 81.38 ± 0.90 | 50.79 ± 0.66 |
| | TabSyn+TCM | 16.86 ± 1.36 | 16.16% ↓ | 76.84 ± 0.34 | 96.16 ± 1.24 | 40.69 ± 2.46 | 98.02 ± 1.62 | 96.51 ± 1.42 | 97.65 ± 0.65 | 51.16 ± 1.82 |
| | TabSyn+TCMP | 17.60 ± 0.28 | 12.48% ↓ | 77.17 ± 0.51 | 97.61 ± 0.27 | 44.46 ± 0.60 | 99.03 ± 0.08 | 96.30 ± 1.48 | 98.16 ± 0.65 | 51.20 ± 0.90 |
| **Adult** | STaSy | 26.02 ± 0.89 | - | 90.54 ± 0.17 | 85.79 ± 7.85 | 34.35 ± 2.46 | 89.14 ± 2.29 | 86.00 ± 2.97 | 51.89 ± 14.87 | 50.46 ± 0.39 |
| | STaSy+Mixup | 24.89 ± 1.30 | 4.37% ↓ | 90.74 ± 0.06 | 90.00 ± 1.91 | 34.24 ± 2.47 | 90.28 ± 1.69 | 87.56 ± 1.06 | 52.61 ± 6.52 | 50.08 ± 0.59 |
| | STaSy+SMOTE | 22.92 ± 3.77 | 11.91% ↓ | 90.50 ± 0.24 | 85.81 ± 11.39 | 32.11 ± 5.13 | 86.91 ± 0.81 | 84.36 ± 2.36 | 45.12 ± 8.82 | 50.46 ± 0.20 |
| | STaSy+TCM | 20.89 ± 1.33 | 19.71% ↓ | 90.45 ± 0.30 | 85.39 ± 1.61 | 31.24 ± 0.97 | 88.33 ± 3.63 | 85.39 ± 4.03 | 45.49 ± 4.78 | 50.92 ± 0.39 |
| | STaSy+TCMP | 21.45 ± 2.60 | 17.59% ↓ | 90.72 ± 0.06 | 86.71 ± 4.12 | 32.63 ± 1.81 | 89.62 ± 1.55 | 86.05 ± 2.44 | 49.12 ± 9.95 | 50.75± 0.59 |
| | TabDDPM | 31.01 ± 0.18 | - | 91.09 ± 0.07 | 93.58 ± 1.99 | 51.52 ± 2.29 | 98.84 ± 0.03 | 97.78 ± 0.07 | 94.63 ± 1.19 | 51.56 ± 0.34 |
| | TabDDPM+Mixup | 30.04 ± 0.41 | 3.14% ↓ | 90.82 ± 0.12 | 95.78 ± 0.68 | 47.65 ± 1.35 | 98.02 ± 1.08 | 96.78 ± 1.33 | 93.65 ± 3.59 | 50.86 ± 0.86 |
| | TabDDPM+SMOTE | 28.98 ± 0.78 | 6.56% ↓ | 90.41 ± 0.36 | 94.93 ± 1.72 | 46.10 ± 0.65 | 93.40 ± 1.12 | 90.76 ± 1.76 | 80.75 ± 0.84 | 51.82 ± 0.56 |
| | TabDDPM+TCM | 27.55 ± 0.19 | 11.16% ↓ | 91.15 ± 0.06 | 94.97 ± 0.06 | 47.43 ± 1.46 | 98.65 ± 0.03 | 97.75 ± 0.07 | 85.61 ± 16.03 | 50.99 ± 0.65 |
| | TabDDPM+TCMP | 26.10 ± 2.11 | 15.83% ↓ | 90.54 ± 0.17 | 92.26 ± 6.97 | 43.49 ± 3.74 | 95.10 ± 4.27 | 91.50 ± 6.53 | 84.76 ± 10.12 | 50.68 ± 0.89 |
| | TabSyn | 29.26 ± 0.23 | - | 91.13 ± 0.09 | 99.31 ± 0.39 | 48.00 ± 0.22 | 99.33 ± 0.09 | 98.19 ± 0.50 | 98.68 ± 0.41 | 50.42 ± 0.27 |
| | TabSyn+Mixup | 28.29 ± 0.28 | 3.30% ↓ | 90.75 ± 0.24 | 98.63 ± 0.81 | 45.73 ± 2.67 | 98.30 ± 0.90 | 97.91 ± 0.12 | 98.05 ± 2.22 | 50.97 ± 1.10 |
| | TabSyn+SMOTE | 27.10 ± 0.15 | 7.36% ↓ | 89.97 ± 0.76 | 98.60 ± 0.50 | 44.72 ± 0.45 | 94.47 ± 0.57 | 91.74 ± 0.42 | 82.55 ± 0.71 | 48.42 ± 0.78 |
| | TabSyn+TCM | 27.03 ± 0.22 | 7.60% ↓ | 91.09 ± 0.17 | 99.04 ± 0.42 | 44.95 ± 0.42 | 99.40 ± 0.07 | 98.51 ± 0.08 | 89.18 ± 1.94 | 50.67 ± 0.11 |
| | TabSyn+TCMP | 25.99 ± 0.52 | 11.17% ↓ | 90.96 ± 0.16 | 98.43 ± 1.04 | 43.23 ± 2.96 | 98.38 ± 0.91 | 96.53 ± 1.47 | 93.39 ± 6.01 | 50.30 ± 0.78 |
| **Shoppers** | STaSy | 25.51 ± 0.32 | - | 91.26 ± 0.23 | 88.02 ± 3.54 | 34.58 ± 1.84 | 88.18 ± 0.29 | 89.10 ± 0.53 | 47.85 ± 8.48 | 51.68 ± 0.56 |
| | STaSy+Mixup | 24.80 ± 1.20 | 2.81% ↓ | 91.79 ± 0.58 | 87.03 ± 5.46 | 38.48 ± 4.54 | 87.14 ± 1.87 | 88.72 ± 1.42 | 47.42 ± 4.84 | 50.36 ± 2.45 |
| | STaSy+SMOTE | 22.52 ± 1.51 | 11.73% ↓ | 91.31 ± 1.21 | 85.22 ± 3.20 | 30.53 ± 1.65 | 81.22 ± 2.23 | 84.74 ± 0.78 | 38.92 ± 2.63 | 46.47 ± 0.95 |
| | STaSy+TCM | 22.78 ± 0.69 | 10.71% ↓ | 90.56 ± 0.44 | 86.66 ± 4.18 | 34.08 ± 1.46 | 87.16 ± 3.78 | 86.56 ± 4.26 | 50.08 ± 6.30 | 50.61 ± 0.41 |
| | STaSy+TCMP | 22.19 ± 1.21 | 13.03% ↓ | 91.37 ± 0.65 | 85.82 ± 2.66 | 34.11 ± 2.08 | 87.38 ± 2.30 | 88.61 ± 1.64 | 52.42 ± 2.65 | 51.19 ± 0.95 |
| | TabDDPM | 31.37 ± 0.31 | - | 92.17 ± 0.32 | 93.16 ± 1.58 | 52.57 ± 1.30 | 97.08 ± 0.46 | 92.92 ± 3.27 | 86.74 ± 0.63 | 51.36 ± 0.63 |
| | TabDDPM+Mixup | 27.45 ± 1.88 | 12.50% ↓ | 91.44 ± 1.37 | 94.80 ± 0.68 | 51.72 ± 1.05 | 92.14 ± 4.16 | 89.31 ± 3.91 | 82.34 ± 3.24 | 46.85 ± 5.81 |
| | TabDDPM+SMOTE | 26.64 ± 1.46 | 15.07% ↓ | 89.96 ± 0.95 | 94.41 ± 4.67 | 45.22 ± 3.26 | 90.78 ± 0.49 | 83.09± 2.47 | 64.05 ± 1.44 | 51.94 ± 1.52 |
| | TabDDPM+TCM | 25.56 ± 1.17 | 18.51% ↓ | 92.17 ± 0.26 | 94.41 ± 1.49 | 50.05 ± 1.59 | 97.18 ± 0.34 | 93.95± 0.51 | 86.96 ± 0.50 | 47.52± 1.81 |
| | TabDDPM+TCMP | 28.51 ± 0.35 | 9.12% ↓ | 92.09 ± 0.99 | 93.43 ± 1.65 | 52.30 ± 0.73 | 97.31 ± 0.22 | 94.79± 0.30 | 87.02 ± 2.04 | 50.83 ± 0.59 |
| | TabSyn | 27.68 ± 0.10 | - | 91.76 ± 0.66 | 99.20 ± 0.29 | 47.79 ± 0.77 | 98.54 ± 0.19 | 97.83 ± 0.10 | 95.44 ± 0.39 | 52.50 ± 0.44 |
| | TabSyn+Mixup | 28.01 ± 0.46 | −1.18% ↓ | 92.02 ± 0.29 | 98.57 ± 0.32 | 48.17 ± 0.84 | 97.59 ± 0.09 | 97.98 ± 0.14 | 98.37 ± 0.47 | 51.50 ± 2.63 |
| | TabSyn+SMOTE | 26.43 ± 0.85 | 4.54% ↓ | 91.96 ± 1.02 | 95.27 ± 0.97 | 44.57 ± 0.24 | 94.58 ± 0.48 | 94.59 ± 0.08 | 79.89 ± 1.22 | 49.99 ± 0.81 |
| | TabSyn+TCM | 25.38 ± 0.18 | 8.30% ↓ | 91.43 ± 0.26 | 99.11 ± 0.28 | 45.98 ± 0.90 | 98.56 ± 0.10 | 97.85 ± 0.06 | 97.28 ± 2.41 | 49.92 ± 1.59 |
| | TabSyn+TCMP | 25.93 ± 0.23 | 6.33% ↓ | 91.75 ± 0.47 | 99.24 ± 0.55 | 46.48 ± 0.77 | 98.60 ± 0.14 | 97.77 ± 0.09 | 97.40 ± 0.57 | 50.21 ± 3.33 |

all fidelity and diversity of synthetic data;[6] 3) downstream tasks performance machine learning efficiency (MLE)[7]. We report AUC in Table 1, i.e., the testing performance (e.g., AUC) on real data when trained only on synthetically generated tabular datasets; 4) C2ST (Classifier Two-Sample Test) evaluates data quality by measuring how well a classifier can distinguish real from synthetic data—lower accuracy suggests better distributional alignment; 5) DCR (Distance to Closest Record) measures privacy risk by quantifying how closely a synthetic sample resembles training vs. holdout samples—lower differences indicate better privacy preserva-

tion. The reported results are averaged over 5 independent experimental runs. More details on evaluation metrics can be found in Appendix D.4.

### 5.2. Memorization and Data Quality: Overall Evaluation

To thoroughly compare the memorization and data generation quality, we incorporate several metrics, including the memorization ratio, MLE, α-precision, β-recall, shape score, and trend score. We report these metrics results of applying TabCutMix and TabCutMixPlus to three SOTA generative models (i.e., STaSy, TabDDPM, and TabSyn) across four datasets in Table. 1. We observe that:

**Obs.1**: TabCutMix and TabCutMixPlus significantly reduce

---

[6]Please see more details on high-order metrics in Appendix D.4.3

[7]Please see more details on MLE in Appendix D.4.2

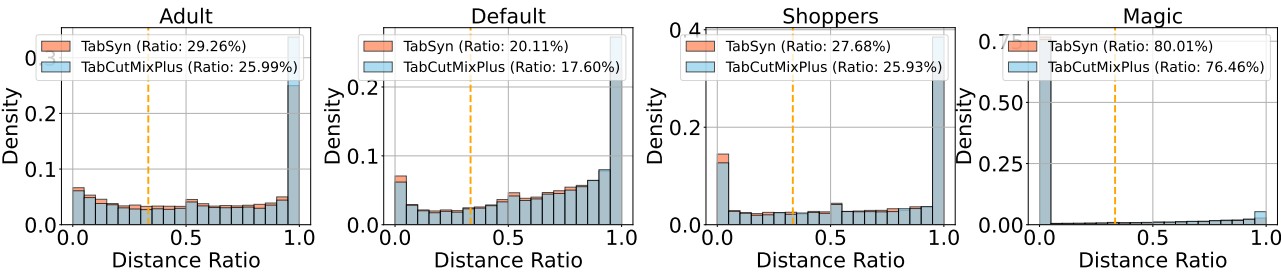

*Figure 4.* The nearest-neighbor distance ratio distributions of TabSyn with and without TabCutMixPlus across different datasets.

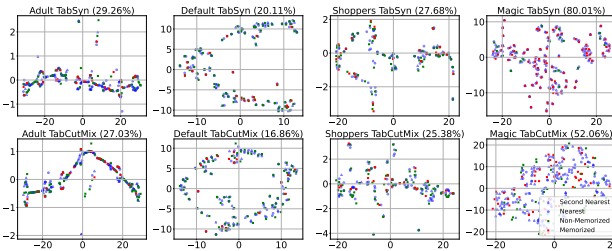

*Figure 5.* The visualization of real and generated samples of TabSyn with and without TabCutMix across different datasets.

the memorization ratio across all models and datasets. For example, in Shoppers dataset, TabCutMix and TabCutMixPlus reduce the memorization ratio by $8.30\%$ and $6.33\%$ for TabSyn model. Although the actual reduction rate varies over dataset and model combination, the overall results indicate that TabCutMix is more effective in mitigating memorization than TabCutMixPlus.

**Obs.2**: TabCutMixPlus demonstrates higher data quality compared to TabCutMix across various metrics, datasets, and diffusion models. For example, in the Default dataset, TabCutMix achieves an MLE of $76.84\%$, $\alpha$-precision of $96.16\%$, while TabCutMixPlus slightly improves it to $77.17\%$ and $97.61\%$ on TabSyn model, suggesting its superior ability to generate high data utility.

### 5.3. A Closer Look at Memorization

#### 5.3.1. DISTANCE RATIO DISTRIBUTION

We analyze the distribution of the nearest-neighbor distance ratio, defined as $r(\boldsymbol{x}) = \frac{\mathrm{NN}_1(\boldsymbol{x}, \mathcal{D})}{\mathrm{NN}_2(\boldsymbol{x}, \mathcal{D})}$, to assess the severity of memorization. A more zero-concentrated ratio distribution indicates a more severe memorization issue, as the generated sample $x$ is closer to a real sample in the training set $\mathcal{D}$. Figure 4 illustrates the distance ratio distribution for both the original TabSyn and TabSyn with TabCutMixPlus, and we observe the following:

**Obs.1**: TabCutMixPlus shifts the distance ratio distribution further away from zero compared to TabSyn, indicating

a further reduction in memorization. For example, in the Magic dataset, TabCutMixPlus reduces the memorization ratio from $80.01\%$ to $76.46\%$, generating samples that are less tightly aligned with the real data $\mathcal{D}$.

**Obs.2**: The distance ratio distributions with TabCutMixPlus exhibit a bipolar pattern, with high probabilities near 0 and 1. However, TabCutMixPlus improves the spread of the distribution by reducing the probability mass near 0 and increasing it near 1. This indicates that TabCutMixPlus better balances memorization reduction and diversity improvement.

#### 5.3.2. VISUALIZATION OF REAL AND GENERATION SAMPLES

We visualize the distribution of real and generative samples for four datasets (i.e., Adult, Default, Shoppers, and Magic) in Figure 5. For each dataset, we sample 100 generated samples while preserving the memorization ratio consistent with that of the entire generated dataset. For each of these 100 samples, we then select their nearest and second-nearest real samples from the training set to visualize. Using t-SNE, we embed both the generative samples and their corresponding nearest and second-nearest real samples from the training data. We make the following observations:

**Obs.1**: In the TabSyn model, memorized generative samples (marked with $\times$) are tightly clustered around their nearest real samples (shown in blue), indicating a high level of memorization. This clustering is particularly pronounced in the Magic dataset, where most generative samples are concentrated near their nearest neighbors, corresponding to a memorization ratio of $80.01\%$. In contrast, non-memorized samples are more dispersed, demonstrating better diversity.

**Obs.2**: While the visual impact of TabCutMix is subtle, we observe that the generative samples exhibit a slightly broader distribution, particularly in datasets like Default and Shoppers. This suggests a reduction in tight clustering around real samples, which correlates with the reduction in memorization ratios. However, in some datasets like Magic, the visual distinction remains modest, indicating that TabCutMix quantitatively reduces memorization.

*Table 2.* The real and generative samples by TabSyn and TabSyn with TabCutMix and TabCutMixPlus in Adult dataset. TCM and TCMP represent TabCutMix and TabCutMixPlus, respectively.

| Samples | Age | Workclass | fnlwgt | Education | Education.num | Marital Status | Occupation | Relationship | Race | Sex | Capital Gain | Capital Loss | Hours per Week | Native Country | Income |
|---|---|---|---|---|---|---|---|---|---|---|---|---|---|---|---|
| Real | 47.0 | Private | 207207.0 | HS-grad | 9.0 | Divorced | Sales | Unmarried | White | Female | 0.0 | 0.0 | 45.0 | United-States | ¡=50K |
| TabSyn | 48.0 | Private | 207915.31 | HS-grad | 9.0 | Divorced | Sales | Unmarried | White | Female | 0.0 | 0.0 | 45.0 | United-States | ¡=50K |
| TabSyn+**TCM** | 36.0 | Private | 201703.6 | HS-grad | 9.0 | Divorced | Sales | Unmarried | White | Female | 0.0 | 0.0 | 60.0 | Germany | ¡=50K |
| Real | 20.0 | Private | 205970.0 | Some-college | 10.0 | Never-married | Craft-repair | Own-child | White | Female | 0.0 | 0.0 | 25.0 | United-States | ¡=50K |
| TabSyn | 19.0 | Private | 208743.81 | Some-college | 10.0 | Never-married | Craft-repair | Own-child | White | Female | 0.0 | 0.0 | 18.0 | United-States | ¡=50K |
| TabSyn+**TCM** | 44.0 | Private | 197128.89 | Some-college | 10.0 | Never-married | Craft-repair | Own-child | White | Female | 0.0 | 0.0 | 40.0 | United-States | ¡=50K |
| Real | 67.0 | Self-emp-not-inc | 106143.0 | Doctorate | 16.0 | Married-civ-spouse | Sales | Husband | White | Male | 20051.0 | 0.0 | 40.0 | United-States | ¿50K |
| TabSyn | 50.0 | Self-emp-not-inc | 151815.17 | Doctorate | 16.0 | Married-civ-spouse | Sales | Husband | White | Male | 15024.0 | 0.0 | 60.0 | United-States | ¿50K |
| TabSyn+**TCM** | 43.0 | Self-emp-not-inc | 250019.6 | Doctorate | 16.0 | Married-civ-spouse | Sales | Husband | White | Male | 0.0 | 1977.0 | 40.0 | United-States | ¿50K |

## 5.4. Case Study on Adult Dataset: Real vs. Generated Samples

Table. 2 provides a comparison between real samples, synthetic samples generated by TabSyn, and synthetic samples generated with TabSyn and TabCutMix (w/ TCM) for the Adult dataset. We report key feature (e.g., age, Workclass, education, marital status, occupation, income, etc.) values of two real samples and the corresponding nearest generative samples to study the quality and characteristics of the generated data.

**Obs.1**: The results 2 suggest that TabSyn alone tends to generate samples that closely resemble real data, raising concerns about memorization. For instance, the top real sample has an age of 47.0 years. TabSyn generates a sample with an age of 48.0 years, which is nearly identical. Similarly, other features like workclass, marital status, and occupation are also closely reproduced.

**Obs.2**: When TabCutMix 2 is applied, the generated age for the top sample changes to 36.0 while the key relationships between other features such as marital status, occupation, and workclass are preserved. For instance, for the workclass feature, all samples across real data, TabSyn, and TabSyn+TCM show "Private," and for the relationship feature, they show "Unmarried" or "Own-child," depending on the context. For the bottom sample, prior to applying TabCutMix, the distance ratio is 0.17, which is less than the threshold of $\frac{1}{3}$ and thus considered memorized. However, after applying TabCutMix, the closest sample achieves a distance ratio of 0.88, significantly exceeding the $\frac{1}{3}$ threshold, indicating a much lower likelihood of memorization. This demonstrates that TabCutMix can introduce diversity in specific features like age while preserving categorical feature relationships.

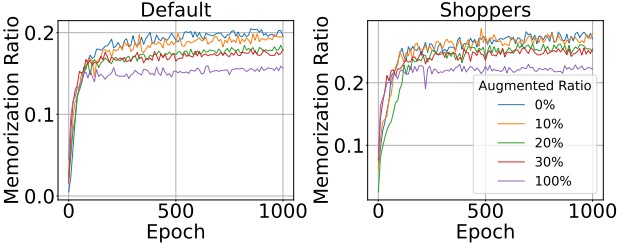

*Figure 6.* The memorization ratio v.s. training epochs with different augmented ratios for TabSyn.

## 5.5. Hyperparameter Study: Impact of Augmented Ratio

In this section, we investigate the effect of the augmented ratio in TabCutMix on the memorization rate. Figure 6 and Table 3 present the memorization ratio for different augmented ratios across two datasets, Default and Shoppers. We test various augmented ratios, including 0%, 10%, 20%, 30%, and 100%, to analyze their impact on the memorization behavior over training epochs.

We observe that the memorization ratio decreases consistently as the augmented ratio increases. Without augmentation (i.e., 0% augmented ratio), the memorization ratio is higher, stabilizing around 20.11% for Default and 27.68% for Shoppers. In contrast, the 100% augmented ratio (purple curve) yields the lowest memorization ratio, stabilizing at approximately 15.34% for Default and 22.06% for Shoppers. This suggests that higher augmented ratios introduce more data diversity, effectively reducing overfitting and preventing the model from memorizing specific training samples.

*Table 3.* Different augmentation ratios (Aug. Ratio) for TabCutMix.

| Aug. Ratio | Default | Shoppers |
|---|---|---|
| 0% | 20.11% | 27.68% |
| 10% | 19.45% | 27.20% |
| 20% | 17.82% | 26.05% |
| 30% | 16.86% | 25.38% |
| 100% | 15.34% | 22.06% |

## 6. Conclusions

In this study, we first investigate memorization phenomena in diffusion models for tabular data using quantitative metrics. Our findings reveal the prevalent memorization behaviors in existing tabular diffusion models, with the memorization ratio increasing as training epochs grow. We further study the effects of the diffusion model instantiation, dataset size, and feature dimensions through the lens of memorization ratio and observe the heterogeneous trend dependent on the dataset. The theoretical analysis provides new insights into why memorization occurs within the SOTA model TabSyn. To address this issue, we propose TabCutMix, which reduces memorization by swapping feature segments between samples, and TabCutMixPlus, which improves upon this by clustering correlated features to preserve feature relationships and address out-of-distribution challenges. Ex-

periments demonstrate that both TabCutMix and TabCut-MixPlus significantly mitigate memorization while maintaining high-quality synthetic data generation. Our work not only highlights the critical issue of memorization in tabular diffusion models but also offers effective solutions with TabCutMix and TabCutMixPlus.

## Impact Statement

This paper presents work whose goal is to advance the field of Machine Learning. There are many potential societal consequences of our work, none which we feel must be specifically highlighted here.

## Acknowledgements

This work is supported in part by NSF CCF-2200255, NSF CCF-2006780, NSF IIS-2027667, NIH U01AG073323, NIH R01HG009658, NIH 1R01HL159170 and NIH 1R01NR02010501. This work also made use of the High Performance Computing Resource in the Core Facility for Advanced Research Computing at Case Western Reserve University.

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

# A. Proposition A.1

For the denoising score matching objective, we have the following result[8]:

**Proposition A.1.** *For empirical denoising score matching objective in Eq. (4) with training data $\{\tilde{z}_n | n = 1, 2, \cdots, N\}$, the optimal score function is given by*

$$
\begin{aligned}
\boldsymbol{s}_\theta^*(\boldsymbol{z}_t, t) = {} & \left( \sum_{n=1}^N \exp\left( -\frac{\|\tilde{\boldsymbol{z}}_n - \boldsymbol{z}_t\|_2^2}{2\sigma^2(t)} \right) \right)^{-1} \\
& \times \sum_{n=1}^N \exp\left( -\frac{\|\tilde{\boldsymbol{z}}_n - \boldsymbol{z}_t\|_2^2}{2\sigma^2(t)} \right) \cdot \frac{\tilde{\boldsymbol{z}}_n - \boldsymbol{z}_t}{\sigma^2(t)}.
\end{aligned}
\tag{6}
$$

Proposition.A.1 provides a closed-form expression for the optimal score matching function given a finite training set. In this section, we prove the close form of optimal score matching function $\boldsymbol{s}_\theta^*(\boldsymbol{z}_t, t)$. Note that the objective of denoising score matching is given by

$$
\min_\theta \mathbb{E}_{\boldsymbol{z}_0 \sim p(\boldsymbol{z}_0)} \mathbb{E}_{\boldsymbol{z}_t \sim p(\boldsymbol{z}_t | \boldsymbol{z}_0)} \| \boldsymbol{s}_\theta(\boldsymbol{z}_t, t) - \nabla_{\boldsymbol{z}_t} \log p(\boldsymbol{z}_t | \boldsymbol{z}_0) \|_2^2,
\tag{7}
$$

Note that the score function can be simplified as

$$
\begin{aligned}
\nabla_{\boldsymbol{z}_t} \log p(\boldsymbol{z}_t | \boldsymbol{z}_0) &= \frac{1}{p(\boldsymbol{z}_t | \boldsymbol{z}_0)} \nabla_{\boldsymbol{z}_t} p(\boldsymbol{z}_t | \boldsymbol{z}_0) \\
&= \frac{1}{p(\boldsymbol{z}_t | \boldsymbol{z}_0)} \cdot \left( -\frac{\boldsymbol{z}_t - \boldsymbol{z}_0}{\sigma^2(t)} \right) \cdot p(\boldsymbol{z}_t | \boldsymbol{z}_0) \\
&= -\frac{1}{\sigma^2(t)} \left( \boldsymbol{z}_0 + \sigma(t)\boldsymbol{\epsilon} - \boldsymbol{z}_0 \right) = -\frac{\boldsymbol{\epsilon}}{\sigma(t)}
\end{aligned}
\tag{8}
$$

Additionally, the noise sample $\boldsymbol{z}_t = \tilde{\boldsymbol{z}}_n + \sigma(t)\boldsymbol{\epsilon}$, we have $\boldsymbol{\epsilon} = -\frac{\tilde{\boldsymbol{z}}_n - \boldsymbol{z}_t}{\sigma(t)}$ and $d\boldsymbol{\epsilon} = \frac{d\boldsymbol{z}_t}{\sigma(t)}$ We can obtain the empirical objective of denoising score matching as follows:

$$
\begin{aligned}
\mathcal{L}_{emp} &= \frac{1}{N} \int \sum_{n=1}^N \left\| \boldsymbol{s}_\theta(\boldsymbol{z}_t, t) + \frac{\boldsymbol{\epsilon}}{\sigma(t)} \right\|_2^2 \mathcal{N}(\boldsymbol{\epsilon}; \boldsymbol{0}, \mathbf{I}) d\boldsymbol{\epsilon} \\
&= \frac{1}{N} \int \sum_{n=1}^N \left\| \boldsymbol{s}_\theta(\boldsymbol{z}_t, t) - \frac{\tilde{\boldsymbol{z}}_n - \boldsymbol{z}_t}{\sigma^2(t)} \right\|_2^2 \mathcal{N}\left( \boldsymbol{z}_t; \tilde{\boldsymbol{z}}_n, \sigma^2(t)\mathbf{I} \right) d\sigma(t) d\boldsymbol{z}_t.
\end{aligned}
\tag{9}
$$

The minimization of empirical loss $\mathcal{L}_{emp}$ is a convex optimization problem. Therefore, the optimum can be obtained via first-order gradient w.r.t. score function $\boldsymbol{s}_\theta(\boldsymbol{z}_t, t)$:

$$
\begin{aligned}
\boldsymbol{0} &= \nabla_{\boldsymbol{s}_\theta(\boldsymbol{z}_t, t)} \left[ \frac{1}{N} \sum_{n=1}^N \left\| \boldsymbol{s}_\theta(\boldsymbol{z}_t, t) - \frac{\tilde{\boldsymbol{z}}_n - \boldsymbol{z}_t}{\sigma^2(t)} \right\|_2^2 \mathcal{N}\left( \boldsymbol{z}_t; \tilde{\boldsymbol{z}}_n, \sigma^2(t)\mathbf{I} \right) \right] \\
&= \frac{2}{N} \sum_{n=1}^N \left[ \boldsymbol{s}_\theta(\boldsymbol{z}_t, t) - \frac{\tilde{\boldsymbol{z}}_n - \boldsymbol{z}_t}{\sigma^2(t)} \right] \mathcal{N}\left( \boldsymbol{z}_t; \tilde{\boldsymbol{z}}_n, \sigma^2(t)\mathbf{I} \right) \\
&= \frac{2}{N} \left\{ \sum_{n=1}^N \mathcal{N}\left( \boldsymbol{z}_t; \tilde{\boldsymbol{z}}_n, \sigma^2(t)\mathbf{I} \right) \boldsymbol{s}_\theta(\boldsymbol{z}_t, t) - \sum_{n=1}^N \mathcal{N}\left( \boldsymbol{z}_t; \tilde{\boldsymbol{z}}_n, \sigma^2(t)\mathbf{I} \right) \frac{\tilde{\boldsymbol{z}}_n - \boldsymbol{z}_t}{\sigma^2(t)} \right\},
\end{aligned}
\tag{10}
$$

---

[8]The analysis is closely related to prior work (Gu et al., 2023) in image generation, where a similar analysis was performed in different generative models. Our work specifically addresses tabular data with mixed feature types by combining a VAE with latent diffusion to handle tabular data.

Therefore, the optimal score function can be written as

$$
\begin{aligned}
s_\theta^*(z_t, t) &= \frac{\sum_{n=1}^N \mathcal{N}(z_t; \tilde{z}_n, \sigma^2(t)\mathbf{I}) \frac{\tilde{z}_n - z_t}{\sigma^2(t)}}{\sum_{n=1}^N \mathcal{N}(z_t; \tilde{z}_n, \sigma^2(t)\mathbf{I})} \\
&= \Big(\sum_{n=1}^N \exp\big(-\frac{\|\tilde{z}_n - z_t\|_2^2}{2\sigma^2(t)}\big)\Big)^{-1} \sum_{n=1}^N \exp\big(-\frac{\|\tilde{z}_n - z_t\|_2^2}{2\sigma^2(t)}\big) \cdot \frac{\tilde{z}_n - z_t}{\sigma^2(t)}
\end{aligned}
\tag{11}
$$

## B. Proof of Proposition. 3.1

Consider the reverse process of a diffusion model defined by the score function $s_\theta(z, t)$ and the following backward stochastic differential equation (SDE):

$$
dz_t = -2\dot{\sigma}(t)\sigma(t)s(z_t, t)dt + \sqrt{2\dot{\sigma}(t)\sigma(t)}d\omega_t,
\tag{12}
$$

where $\omega_t$ is standard Brownian motion, and $\sigma(t)$ are noise ratio at time instant $t$.

For solving this backward SDE given optimal score function $s_\theta^*(z_t, t)$, we consider the following steps:

**Step 1: Euler Approximation.** We use Euler approximation for backward SDE via sampling multiple time steps $0 = t_0 < t_1 = \tau < t_2 = 2\tau < \cdots < t_n = n\tau = T$, where $\tau$ is time sampling resolution and small value indicates low approximation error. Using an Euler discretization, the backward SDE can be approximated at discrete time steps $t_n$, leading to the following update rule:

$$
z_{t_n} = z_{t_{n+1}} - 2\dot{\sigma}(t)\sigma(t)\Big\|_{t=t_{n+1}} s(z_t, t)(t_n - t_{n+1}) + \sqrt{2\dot{\sigma}(t)\sigma(t)}\|_{t=t_{n+1}} \cdot \epsilon \cdot (t_n - t_{n+1}),
\tag{13}
$$

**Step 2: Update Rule Calculation.** Next, we calculate the update rule considering infinite short time resolution $\tau \to 0$,

$$
\lim_{t_n - t_{n+1} \to 0^-} 2\dot{\sigma}(t)\sigma(t)\Big\|_{t=t_{n+1}} = 2\sigma(t_{n+1})\frac{\sigma(t_n) - \sigma(t_{n+1})}{t_n - t_{n+1}},
\tag{14}
$$

then we have

$$
\begin{aligned}
z_{t_n} &= z_{t_{n+1}} - 2\sigma(t_{n+1})\big(\sigma(t_n) - \sigma(t_{n+1})\big)s(z_t, t) \\
&\quad + \sqrt{2\sigma(t_{n+1})\big(\sigma(t_n) - \sigma(t_{n+1})\big)(t_n - t_{n+1})} \cdot \epsilon,
\end{aligned}
\tag{15}
$$

For $t_0 = 0$, it is easy to obtain $\sigma(t) = 0$, the generated sample in latent space $z_0$ is giving by

$$
z_0 = z_\tau + 2\sigma^2(\tau)s(z_t, t) + \sqrt{2\tau\sigma^2(\tau)} \cdot \epsilon.
\tag{16}
$$

**Step 3: The generated sample in latent space under $\tau \to 0$.** When the denoising score function perfectly approximates the optimal solution, we have

$$
s(z_t, t) = s_\theta^*(z_t, t) = \Big(\sum_{n=1}^N \exp\big(-\frac{\|\tilde{z}_n - z_t\|_2^2}{2\sigma^2(t)}\big)\Big)^{-1} \sum_{n=1}^N \exp\big(-\frac{\|\tilde{z}_n - z_t\|_2^2}{2\sigma^2(t)}\big) \cdot \frac{\tilde{z}_n - z_t}{\sigma^2(t)},
\tag{17}
$$

Subsequently, we consider the optimal score function under $\tau \to 0$. Suppose the nearest neighbor of $z$ is $\tilde{z}_m = \text{NN}_1(z, \mathcal{D})$, we have

$$
\|z - \tilde{z}_m\|_2^2 - \|z - \tilde{z}_n\|_2^2 < 0, \quad \text{for} \quad n \neq m.
\tag{18}
$$

Define distribution:

$$
p_t(z = \tilde{z}_n) = \Big(\sum_{n=1}^N \exp\big(-\frac{\|\tilde{z}_n - z_t\|_2^2}{2\sigma^2(t)}\big)\Big)^{-1} \sum_{n=1}^N \exp\big(-\frac{\|\tilde{z}_n - z_t\|_2^2}{2\sigma^2(t)}\big),
\tag{19}
$$

where $n = 1, 2, \cdots, N$. Note that $\sigma(\tau) \to 0$ if $\tau \to 0$. It is easy to calculate

$$
\begin{aligned}
\lim_{\tau \to 0} p_\tau(\boldsymbol{z} = \tilde{\boldsymbol{z}}_m) &= \lim_{\tau \to 0} \Big( \sum_{n=1}^{N} \exp \big( - \frac{\|\tilde{\boldsymbol{z}}_m - \boldsymbol{z}_\tau\|_2^2}{2\sigma^2(\tau)} \big) \Big)^{-1} \sum_{n=1}^{N} \exp \big( - \frac{\|\tilde{\boldsymbol{z}}_m - \boldsymbol{z}_\tau\|_2^2}{2\sigma^2(\tau)} \big) \\
&= \lim_{\tau \to 0} \Big[ 1 + \sum_{n \neq m} \exp \big( - \frac{\|\tilde{\boldsymbol{z}}_m - \boldsymbol{z}_\tau\|_2^2}{2\sigma^2(\tau)} \big) \Big]^{-1} \\
&= \Big[ 1 + \lim_{\sigma(\tau) \to 0} \sum_{n \neq m} \exp \big( - \frac{\|\tilde{\boldsymbol{z}}_m - \boldsymbol{z}_\tau\|_2^2}{2\sigma^2(\tau)} \big) \Big]^{-1} = 1,
\end{aligned} \tag{20}
$$

similarly, we have, for any $n' \neq m$,

$$
\lim_{\tau \to 0} p_\tau(\boldsymbol{z} = \tilde{\boldsymbol{z}}_{n'}) = 0. \tag{21}
$$

According to the above equations, the optimal score function is given by

$$
\lim_{\tau \to 0} \boldsymbol{s}_\theta^*(\boldsymbol{z}_t, t) = \frac{\tilde{\boldsymbol{z}}_m - \boldsymbol{z}_t}{\sigma^2(t)}, \tag{22}
$$

and the generated sample in latent space $\boldsymbol{z}_0$ is as follows:

$$
\begin{aligned}
\lim_{\tau \to 0} \boldsymbol{z}_0 &= \lim_{\tau \to 0} \boldsymbol{z}_\tau + 2\sigma^2(\tau)\boldsymbol{s}(\boldsymbol{z}_t, t) + \sqrt{2\tau\sigma^2(\tau)} \cdot \boldsymbol{\epsilon} \\
&= \lim_{\tau \to 0} \boldsymbol{z}_\tau + 2\sigma^2(\tau) \frac{\tilde{\boldsymbol{z}}_m - \boldsymbol{z}_t}{\sigma^2(t)} + \sqrt{2\tau\sigma^2(\tau)} \cdot \boldsymbol{\epsilon} \\
&= 2\tilde{\boldsymbol{z}}_m - \lim_{\tau \to 0} \boldsymbol{z}_\tau,
\end{aligned} \tag{23}
$$

Therefore, we have $\lim_{\tau \to 0} \boldsymbol{z}_0 = \tilde{\boldsymbol{z}}_m = \mathrm{NN}_1(\boldsymbol{z}_\tau, \mathcal{D})$.

To summarize, under the assumption (1) the neural network can perfectly approximate the score function $\boldsymbol{s}(\boldsymbol{z}_t, t) = \boldsymbol{s}_\theta^*(\boldsymbol{z}_t, t)$ (2) perfect SDE solver with infinite time solution ($\tau \to 0$), the generated sample $\boldsymbol{z}_0$ replicates one of the training samples from the dataset $\mathcal{D}$.

## C. Algorithm

In this section, we provide an algorithmic illustration of the proposed TabCutMix and TabCutMixPlus in Algorithms 1 and 2, respectively. In TabCutMix/TabCutMixPlus, the hyperparameter $r_n$ determines the augmentation ratio, i.e., the number of augmented samples over the whole number of the original training samples.

---

**Algorithm 1** Pseudo-code of TabCutMix

---

**Require:** Training set $\mathcal{D}$, Number of samples $N$
1: Augmented sample set $\tilde{\mathcal{D}} = \emptyset$
2: **for** $i = 1$ to $N$ **do**
3:     Sample class $c$ from $\{1, \cdots, C\}$ with prior class distribution;          ▷ Keep class ratio after augmentation.
4:     Sample $(\boldsymbol{x}_A, y_A)$ and $(\boldsymbol{x}_B, y_B)$ from class $c$ in $\mathcal{D}$;    ▷ Randomly select two training samples from the same class.
5:     Sample $\lambda \sim \mathrm{Unif}(0, 1)$ and sampling binary mask $M$ with Bernoulli distribution $\mathrm{Bern}(\lambda)$; ▷ Proportion of features to exchange.
6:     $\tilde{\boldsymbol{x}} \leftarrow M \odot \boldsymbol{x}_A + (1 - M) \odot \boldsymbol{x}_B$;                 ▷ Mix the features based on binary mask $M$.
7:     $\tilde{y} \leftarrow c$;                                             ▷ Assign the label of the new sample.
8:     $\tilde{\mathcal{D}} = \tilde{\mathcal{D}} \cup (\tilde{\boldsymbol{x}}, \tilde{y})$;                                ▷ Save the augmented sample.
9: **end for**
10: **return** New Training Set $\mathcal{D} \cup \tilde{\mathcal{D}}$

---

---

**Algorithm 2** Pseudo-code of TabCutMixPlus

---

**Require:** Training set $\mathcal{D}$, Number of samples $N$
1:  Augmented sample set $\tilde{\mathcal{D}} = \emptyset$
2:  Calculate correlation metrics for features: (a) Pearson correlation coefficient for numerical feature; (b) Cramér's V based on contingency tables for categorical features; (c) ETA coefficient for numerical-categorical pairs.
3:  Perform hierarchical clustering on features using correlation metrics;        ▷ Group features based on similarity.
4:  **for** $i = 1$ to $N$ **do**
5:       Sample class $c$ from $\{1, \cdots, C\}$ with prior class distribution;        ▷ Keep class ratio after augmentation.
6:       Sample $(\boldsymbol{x}_A, y_A)$ and $(\boldsymbol{x}_B, y_B)$ from class $c$ in $\mathcal{D}$;     ▷ Randomly select two training samples from the same class.
7:       **for** each cluster $k$ **do**
8:           Sample $\lambda \sim \text{Unif}(0,1)$ and sampling binary mask $M_k$ with Bernoulli distribution $\text{Bern}(\lambda)$;        ▷ Proportion of features to exchange within cluster $k$.
9:           $\tilde{\boldsymbol{x}}_k \leftarrow M_k \odot \boldsymbol{x}_{A,k} + (1 - M_k) \odot \boldsymbol{x}_{B,k}$;        ▷ Mix features in cluster $k$ based on binary mask $M_k$.
10:           Add $\tilde{\boldsymbol{x}}_k$ to $\tilde{\boldsymbol{x}}$;
11:       **end for**
12:       $\tilde{y} \leftarrow c$;        ▷ Assign the label of the new sample.
13:       $\tilde{\mathcal{D}} = \tilde{\mathcal{D}} \cup (\tilde{\boldsymbol{x}}, \tilde{y})$;        ▷ Save the augmented sample.
14:  **end for**
15:  **return** New Training Set $\mathcal{D} \cup \tilde{\mathcal{D}}$

---

# D. Experimental Details

We implement TabCutMix and all the baseline methods with PyTorch. All the methods are optimized with Adam optimizer.

## D.1. Datasets

We select 7 datasets, 5 of 7 datasets come from UCI Machine Learning Repository: Adult, Default, Shoppers, Magic, and Wilt. The other two are Cardio and Churn Modeling. All datasets are associated with classification tasks.

The statistics are shown in Table. 4. The detailed introduction for these datasets are given as follows:

- **Adult Dataset**[9]: The Adult Census Income dataset consists of demographic and employment-related information about individuals, derived from the 1994 U.S. Census. The dataset's primary task is to predict whether an individual earns more or less than $50,000$ per year. It includes features such as age, education, work class, marital status, and occupation, with $48,842$ records. This dataset is widely used in binary classification tasks, especially for exploring income prediction and socio-economic factors.

- **Default Dataset**[10]: The Default of Credit Card Clients Dataset contains records of default payments, credit history, demographic factors, and bill statements of credit card holders in Taiwan, covering data from April 2005 to September 2005. It features $30,000$ clients and aims to predict whether a client will default on payment the following month. Key features include credit limit, past payment status, and monthly bill amounts, making it useful for credit risk modeling and financial behavior analysis.

- **Shoppers Dataset**[11]: The Online Shoppers Purchasing Intention Dataset includes detailed information about user interactions with online shopping websites, with data from $12,330$ user sessions. It records features such as the number of pages viewed, time spent on different sections of the site, and user behavior metrics. The primary task is to predict whether a user's session will result in a purchase. This dataset is particularly useful for studying customer behavior, e-commerce optimization, and purchase prediction models.

- **Magic Dataset**[12]: The Magic Gamma Telescope Dataset is designed for the classification of high-energy gamma particles collected by a ground-based atmospheric Cherenkov telescope. The dataset contains $19,019$ instances and is used to distinguish between signals from gamma particles and background noise generated by hadrons. The features include

---

[9] https://archive.ics.uci.edu/dataset/2/adult
[10] https://archive.ics.uci.edu/dataset/350/default+of+credit+card+clients
[11] https://archive.ics.uci.edu/dataset/468/online+shoppers+purchasing+intention+dataset
[12] https://archive.ics.uci.edu/dataset/159/magic+gamma+telescope

statistical properties of the events such as length, width, and energy distribution, making it useful for astronomical data analysis and high-energy particle research.

- **Wilt Dataset**[13]: The Wilt dataset is a high-resolution remote sensing dataset used for binary classification tasks, focusing on detecting diseased trees ('w') versus other land cover ('n'). It includes $4,889$ instances. Features include spectral and texture information derived from Quickbird imagery, such as GLCM mean texture, mean green, red, NIR values, and standard deviation of the Pan band. The dataset is imbalanced, with only 74 samples of diseased trees.

- **Cardio Dataset**[14]: The Cardiovascular Disease dataset consists of $70,000$ patient records, featuring 11 attributes and a binary target variable indicating the presence or absence of cardiovascular disease. The attributes are categorized into three types: *objective* (e.g., age, height, weight, gender), *examination* (e.g., blood pressure, cholesterol, glucose), and *subjective* (e.g., smoking, alcohol intake, physical activity).

- **Churn Modeling Dataset**[15]: The Churn Modeling dataset contains data on $10,000$ customers from a bank, with the target variable indicating whether a customer has churned (closed their account) or not. The dataset includes 14 columns that represent various features such as customer demographics (e.g., age, gender, and geography), account details (e.g., balance, number of products, tenure), and behaviors (e.g., credit score, activity, and churn status).

*Table 4.* Statistics of datasets. *Num* indicates the number of numerical columns, and *Cat* indicates the number of categorical columns.

| Dataset | #Rows | #Num | #Cat | #Train | #Validation | #Test | Task |
|---------|-------|------|------|--------|-------------|-------|------|
| Adult | 48,842 | 6 | 9 | 28,943 | 3,618 | 16,281 | Classification |
| Default | 30,000 | 14 | 11 | 24,000 | 3,000 | 3,000 | Classification |
| Shoppers | 12,330 | 10 | 8 | 9,864 | 1,233 | 1,233 | Classification |
| Magic | 19,019 | 10 | 1 | 15,215 | 1,902 | 1,902 | Classification |
| Cardio | 70,000 | 5 | 7 | 44,800 | 11,200 | 14,000 | Classification |
| Churn Modeling | 10,000 | 7 | 5 | 6,400 | 1,600 | 2,000 | Classification |
| Wilt | 4,839 | 5 | 1 | 3,096 | 775 | 968 | Classification |

In Table 4, the column "# Rows" represents the number of records in each dataset, while "# Num" and "# Cat" indicate the number of numerical and categorical features (including the target feature), respectively. Each dataset is split into training, validation, and testing sets for machine learning efficiency experiments. For the Adult dataset, which has an official test set, we directly use it for testing, while the training set is split into training and validation sets in a ratio of $8:1$. For the remaining datasets, the data is split into training, validation, and test sets with a ratio of 8:1:1, ensuring consistent splitting with a fixed random seed.

### D.2. Alternative Models

In this section, we present and compare the characteristics of the baseline methods employed in this study.

- **CTGAN** (Xu et al., 2019) is a generative model designed specifically for synthetic tabular data generation using a GAN-based framework. CTGAN employs mode-specific normalization to effectively handle numerical columns with complex distributions, ensuring better learning of their patterns. Additionally, it incorporates conditional generation to address imbalances in categorical features by conditioning on specific class distributions, which improves the diversity and utility of the generated data.

- **TVAE** (Xu et al., 2019) is a VAE-based approach tailored for synthetic tabular data generation. Like CTGAN, it uses mode-specific normalization for numerical features and conditional generation for categorical features, but relies on the VAE framework to model data. This approach allows TVAE to capture latent relationships in tabular datasets while addressing challenges like class imbalance and mixed-type data.

- **STaSy** (Kim et al., 2023) is a recently developed diffusion-based model designed for synthetic tabular data generation. It treats one-hot encoded categorical columns as continuous features, allowing them to be processed alongside numerical columns. STaSy utilizes the VP/VE stochastic differential equations (SDEs) to model the distribution of tabular data.

---

[13]https://archive.ics.uci.edu/dataset/285/wilt
[14]https://www.kaggle.com/datasets/sulianova/cardiovascular-disease-dataset
[15]https://www.kaggle.com/datasets/shrutimechlearn/churn-modelling?resource=download

Additionally, the model introduces several training strategies, such as self-paced learning and fine-tuning, to stabilize the training process, thereby improving both the quality and diversity of the generated data.

- **TabDDPM** (Kotelnikov et al., 2023) follows a similar framework to CoDi by applying diffusion models to both numerical and categorical data. Like CoDi, it uses DDPM with Gaussian noise for numerical columns and multinomial diffusion for categorical data. However, TabDDPM simplifies the modeling process by concatenating both numerical and categorical features as inputs to a denoising function, which is implemented as a multi-layer perceptron (MLP). While CoDi incorporates more advanced techniques like inter-conditioning and contrastive learning, TabDDPM's more streamlined approach has been shown to outperform CoDi in experimental evaluations, proving that simplicity can sometimes yield better results.

- **TabSyn** (Zhang et al., 2023a) is a SOTA approach for generating high-quality synthetic tabular data by leveraging diffusion models in a unified latent space. Unlike previous methods that struggle to handle mixed data types, such as numerical and categorical features, TabSyn first transforms raw tabular data into a continuous latent space, where diffusion models with Gaussian noise can be effectively applied. To maintain the underlying relationships between columns, TabSyn uses a Variational AutoEncoder (VAE) architecture that captures both inter-column dependencies and token-level representations. The method employs an adaptive loss weighting technique to fine-tune the balance between reconstruction performance and smooth embedding generation. TabSyn's diffusion process is simplified with Gaussian noise that progressively reduces as the reverse

## D.3. Baselines

In this section, we describe and compare the data augmentation baselines employed in this study: SMOTE, Mixup, and Independent Joint Family (IJF).

- SMOTE: SMOTE (Synthetic Minority Oversampling Technique) (Chawla et al., 2002) is a widely-used oversampling method designed for numerical data augmentation. It generates synthetic samples by interpolating between existing data points within the same class. While effective for numerical features, SMOTE is not designed to handle categorical features directly, which may limit its application in mixed-type tabular datasets.

- Mixup: Mixup (Zhang, 2017) is a data augmentation technique that creates new samples by taking a convex combination of two existing samples and their labels. While Mixup is straightforward and effective for enhancing data diversity, it assumes linear relationships between features, which might not hold true in tabular data. Additionally, Mixup can struggle with preserving the inherent relationships between numerical and categorical features.

- Independent Joint Family (IJF): IJF is a simple augmentation method based on independent feature assumption. Unlike generative models like GANs or VAEs, which are computationally expensive and often unsuitable for generating single-dimensional features, IJF estimates the parameter distribution for numerical features and uses empirical frequency distributions for categorical features. This approach assumes independence between features during augmentation, allowing for efficient and flexible sample generation. The default augmentation ratio for IJF is set to $30\%$, providing a balanced trade-off between data diversity and computational overhead.

## D.4. Evaluation Metrics

### D.4.1. LOW-ORDER STATISTICS

In this part, we will introduce the details of the shape score and trend score[16] for each feature and feature pair, respectively.

The Shape Score of numerical and categorical features are determined by the KSComplement and TVComplement metrics in SDMetrics package, respectively. KSComplement compares the shapes of real and synthetic distributions using the maximum difference between their cumulative distribution function (CDFs). TVComplement is based on the TVComplement, which assesses how well the categorical distributions in the real and synthetic datasets align, with smaller differences leading to a higher score.

- **Shape Score of Numerical Features:** The KSComplement is computed based on the Kolmogorov-Smirnov (KS) statistic. The KS statistic quantifies the maximum distance between the Cumulative Distribution Functions (CDFs) of real and

---

[16]We calculate these scores based on SDMetrics package, available at https://docs.sdv.dev/sdmetrics.

synthetic data distributions. The formula is given by:

$$KST = \sup_x |F_r(x) - F_s(x)|, \tag{24}$$

where $F_r(x)$ and $F_s(x)$ are the CDFs of the real distribution $p_r(x)$ and the synthetic distribution $p_s(x)$, respectively. To ensure that a higher score represents higher quality, we use KSComplement based on shape score $= 1 - KST$. A higher shape score indicates greater similarity between the real and synthetic data distributions, resulting in a higher Shape Score.

- **Shape Score of Categorical Features:** The TVComplement is calculated derived from the Total Variation Distance (TVD). The TVD measures the difference between the probabilities of categorical values in the real and synthetic datasets. It is defined as:

$$TVD = \frac{1}{2} \sum_{\omega \in \Omega} |R(\omega) - S(\omega)|, \tag{25}$$

where $\Omega$ represents the set of all possible categories, and $R(\omega)$ and $S(\omega)$ denote the real and synthetic frequencies for each category. The shape score is defined as shape score $= 1 - TVD$, which returns a score where higher values reflect a smaller difference between real and synthetic category distributions.

In this paper, we report the average shape score across all numerical and categorical features.

The Trend Score is used to evaluate how well the synthetic data captures the relationships between column pairs in the real dataset. Different metrics are applied depending on the types of columns involved: numerical, categorical, or a combination of both.

- **Numerical-Numerical Pairs.** For numerical column pairs, the *Pearson Correlation Coefficient* is used to measure the linear correlation between the two columns. The Pearson correlation, $\rho(x, y)$, is defined as:

$$\rho_{x,y} = \frac{Cov(x, y)}{\sigma_x \sigma_y}, \tag{26}$$

where $Cov(x, y)$ is the covariance, and $\sigma_x$ and $\sigma_y$ are the standard deviations of columns $x$ and $y$, respectively. The trend score for numerical-numerical pair (i.e., correlation similarity) is calculated as 1 minus the average absolute difference between the real data's and synthetic data's correlation values:

$$\text{Trend Score} = 1 - \frac{1}{2}\mathbb{E}_{x,y}\left[|\rho^R(x, y) - \rho^S(x, y)|\right], \tag{27}$$

where $\rho^R(x, y)$ and $\rho^S(x, y)$ denote the Pearson correlation coefficients of the real and synthetic datasets, respectively.

- **Categorical-Categorical Pairs.** For categorical column pairs, the *Contingency Similarity* metric is used. This metric measures the difference between real and synthetic contingency tables using the Total Variation Distance (TVD). The contingency score is defined as:

$$\text{Contingency Score} = \frac{1}{2} \sum_{\alpha \in A} \sum_{\beta \in B} |R_{\alpha,\beta} - S_{\alpha,\beta}|, \tag{28}$$

where $A$ and $B$ are the sets of all possible categories in the two columns, and $R_{\alpha,\beta}$ and $S_{\alpha,\beta}$ represent the joint frequencies of category combinations $\alpha$ and $\beta$ for real and synthetic data, respectively. The trend score is calculated as $1 - \text{Contingency Score}$.

- **Mixed Pairs (Numerical-Categorical).** For column pairs involving one numerical and one categorical column, the numerical column is first discretized into bins. After discretization, the contingency similarity metric is applied to evaluate the relationship between the binned numerical data and the categorical column, similar to how it is used for categorical-categorical pairs. The trend score for mixed pair is calculated as $1 - \text{Contingency Score}$.

Finally, the **Trend Score** is computed as the average of all pairwise scores (Pearson Score for numerical-numerical pairs, and Contingency Score for categorical-categorical and numerical-categorical pairs). This score reflects how well the synthetic data captures the relationships and trends between columns in the real dataset.

D.4.2. MACHINE LEARNING EFFICIENCY EVALUATION

We follow the experimental setting in work (Zhang et al., 2023a). We split each dataset into training and testing sets. The generative models are trained using the real training data, and subsequently, a synthetic dataset of equal size is generated for further experimentation.

To assess the quality of synthetic data in Machine Learning Efficiency (MLE) tasks, we evaluate the divergence in performance when models are trained on either real or synthetic data. The procedure follows these steps: First, the machine learning model is trained using real data, which is split into training and validation sets in an 8:1 ratio. The classifier or regressor is trained on this data, and hyperparameters are optimized based on validation performance. Once the optimal hyperparameters are determined, the model is retrained on the complete training set and evaluated using the real test data. The synthetic data undergoes the same evaluation procedure to assess its impact on model performance.

The following lists the hyperparameter search space for the XGBoost classifier applied during the MLE tasks, where grid search is used to determine the best parameter configurations:

- **Number of estimators:** $\{10, 50, 100\}$
- **Minimum child weight:** $\{5, 10, 20\}$
- **Maximum tree depth:** $\{1, 10\}$
- **Gamma:** $\{0.0, 1.0\}$

The implementations of these evaluation metrics are sourced from SDMetrics[17], and we follow their guidelines for ensuring consistency across real and synthetic data assessments.

D.4.3. SAMPLE-LEVEL QUALITY METRICS: $\alpha$-PRECISION AND $\beta$-RECALL

To rigorously evaluate the quality of synthetic data, we employ two complementary metrics proposed in work (Alaa et al., 2022): $\alpha$-Precision and $\beta$-Recall. These metrics offer a refined approach to assessing the fidelity and diversity of synthetic data samples by focusing on their relationship with the real data distribution.

- **$\alpha$-Precision.** The $\alpha$-Precision metric quantifies the fidelity of synthetic data by measuring the probability that a generated sample lies within the $\alpha$-support of the real data distribution, denoted as $S_r^\alpha$. The $\alpha$-support includes the most representative regions of the real data, containing the highest probability mass. Therefore, a high $\alpha$-Precision score ensures that the synthetic samples are realistic, falling within these high-density areas of the real data. This metric is particularly important because it distinguishes between synthetic samples that resemble real data in a typical way and those that might still be valid but are more akin to outliers. By focusing on the high-density areas, $\alpha$-Precision ensures that the generated data looks both realistic and "typical" compared to real-world data. Mathematically, this is expressed as:

$$P_\alpha = \mathbb{P}(\tilde{X}_g \in S_r^\alpha), \quad \alpha \in [0, 1]. \tag{29}$$

- **$\beta$-Recall.** Conversely, $\beta$-Recall evaluates the coverage of synthetic data. It measures whether the synthetic data captures the entire real data distribution, particularly focusing on the $\beta$-support of the generative model, denoted as $S_g^\beta$. The $\beta$-support includes all regions of the real distribution, not just the frequent or typical areas. A high $\beta$-Recall score indicates that the synthetic data can represent even the rare or low-density parts of the real distribution. This metric is crucial because it ensures that the synthetic data does not merely replicate the most common patterns but also spans the broader diversity of the real data, capturing rare or edge cases. Mathematically, it is defined as:

$$R_\beta = \mathbb{P}(\tilde{X}_r \in S_g^\beta), \quad \beta \in [0, 1]. \tag{30}$$

**Importance of $\alpha$-Precision and $\beta$-Recall.** The combination of $\alpha$-Precision and $\beta$-Recall allows for a holistic assessment of synthetic data. While $\alpha$-Precision ensures that the synthetic data aligns well with the most typical regions of the real data distribution (fidelity), $\beta$-Recall ensures that the synthetic data covers the full diversity of the real data (coverage). Together, these metrics provide insight into both the accuracy and diversity of the synthetic data. By sweeping through values of

---

[17]https://docs.sdv.dev/sdmetrics

$\alpha$ and $\beta$, one can gain a more dynamic understanding of how synthetic data aligns with different aspects of the real data distribution, offering a comprehensive evaluation of its quality.

In summary, $\alpha$-Precision ensures the generated data looks realistic and falls within typical regions of the real distribution, while $\beta$-Recall ensures that the generated data covers the entire distribution, including rare cases. The complementary nature of these two metrics makes them essential for evaluating the fidelity and diversity of synthetic data.

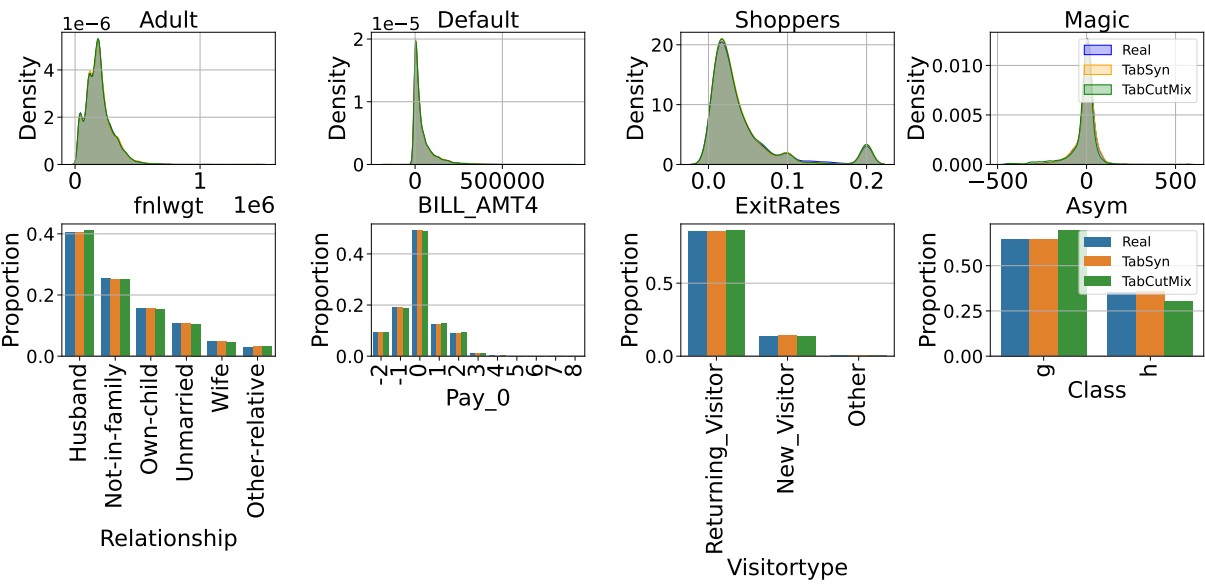

*Figure 7.* Visualization of synthetic data's single column distribution density v.s. the real data.

### D.4.4. DISTANCE TO CLOSEST RECORD (DCR) SCORE

The Distance to the Closest Record (DCR) score is a commonly used metric for assessing privacy leakage risks in synthetic data. This metric quantifies how similar a synthetic sample is to records in the training set compared to those in a holdout set. By calculating the DCR score for each synthetic sample against both the training and holdout sets, we can determine whether the synthetic data poses privacy concerns. If privacy risks are present, DCR scores for the training set would tend to be significantly lower than those for the holdout set, indicating potential memorization of training data. In contrast, the absence of such risks would result in overlapping distributions of DCR scores between the training and holdout sets. Moreover, a probability close to 50% that a synthetic sample is closer to the training set than the holdout set reflects a lack of systematic bias toward the training set, which is a positive indicator for privacy preservation.

Following (Zhang et al., 2023a), employ a "synthetic vs. holdout" evaluation protocol. The dataset is split evenly into two parts: one serves as the training set for the generative model, while the other acts as the holdout set and is excluded from training. After generating a synthetic dataset of the same size as the training and holdout sets, we calculate DCR scores for synthetic samples.

### D.4.5. CLASSIFIER TWO SAMPLE TESTS (C2ST)

The Classifier Two-Sample Test (C2ST) (Zhang et al., 2023a) is used to evaluate how well synthetic data replicates the distribution of real data. This approach involves training a binary classifier to distinguish between real and synthetic samples. If the synthetic data closely matches the distribution of the real data, the classifier should struggle to differentiate the two, resulting in a test accuracy close to 50%. Conversely, if the synthetic data deviates significantly from the real data distribution, the classifier will achieve higher accuracy, indicating poor alignment. The C2ST score provides a quantitative measure of this alignment, offering insights into the quality of the synthetic data. A low C2ST score suggests that the synthetic data effectively captures the real data distribution, making it difficult for the classifier to distinguish between real and synthetic samples.

D.4.6. OUT-OF-DISTRIBUTION (OOD) DETECTION

TabCutMix may introduce a degree of OOD (Yang et al., 2024a) issues. To investigate the potential relationship between TabCutMix and OOD, we conducted OOD detection experiments. These experiments also aimed to evaluate whether TabCutMixPlus could mitigate OOD-related challenges to some extent. We framed the OOD detection task as a classification problem, treating normal samples as negative and OOD samples as positive. Since our dataset lacks explicit labels for OOD samples, we synthesized positive samples following the approach outlined in (Ulmer et al., 2020). For numerical features, we randomly selected one feature and scaled it by a factor $F$ (where $F = 100$). This approach aligns with the methodology in (Azizmalayeri et al., 2023), which experimented with $F$ values of 10, 100, and 1000; we adopted $F = 100$ as a balanced choice for our experiments. For categorical features, we randomly selected a value from the existing categories of the chosen feature. This process was repeated for a single feature at a time. We used the original training set as the negative class and the synthesized samples as the positive class. A multi-layer perceptron (MLP) was trained to classify between these two classes. Subsequently, we tested the samples generated by TabCutMix and TabCutMixPlus using the trained MLP and calculated the proportion of samples classified as OOD. This analysis provides insights into the extent of OOD issues introduced by TabCutMix and the potential of TabCutMixPlus to alleviate such issues.

5 indicates that the OOD issue introduced by TabCutMix is relatively minor across most datasets, as evidenced by low OOD ratios (e.g., 2.06% for **Adult** and 0.61% for **Magic**) and high F1 scores (e.g., above 90% in several cases). While some datasets, such as **Default** and **Cardio**, exhibit higher OOD ratios (39.47% and 4.83%, respectively). TabCutMixPlus significantly mitigates the OOD problem, reducing the OOD ratio by substantial margins across all datasets. For instance, in the **Adult** dataset, the OOD ratio is reduced from 2.06% to 0.36%, while in the **Default** dataset, it decreases from 39.47% to 25.44%. These findings highlight the effectiveness of TabCutMixPlus in addressing potential OOD challenges while maintaining robust classification capabilities, reinforcing its utility in synthetic data augmentation workflows.

*Table 5.* OOD detection of datasets.

| Method | Adult | Default | Shoppers | Magic | Cardio | Churn Modeling | Wilt | Avg. Rank |
|---|---|---|---|---|---|---|---|---|
| Mixup (F1 Score%) | $92.32 \pm 0.67$ | $71.64 \pm 0.74$ | $83.30 \pm 0.77$ | $99.67 \pm 0.04$ | $60.32 \pm 0.47$ | $98.01 \pm 0.07$ | $99.94 \pm 0.01$ | 1.86 |
| Mixup (Ratio%) | $0.03 \pm 0.06$ | $5.19 \pm 2.49$ | $0.16 \pm 0.26$ | $0.00 \pm 0.00$ | $1.98 \pm 0.75$ | $0.00 \pm 0.00$ | $0.00 \pm 0.00$ | 1.14 |
| SMOTE (F1 Score%) | $92.24 \pm 0.79$ | $71.55 \pm 0.78$ | $83.16 \pm 1.03$ | $99.68 \pm 0.05$ | $60.30 \pm 0.49$ | $98.01 \pm 0.09$ | $99.95 \pm 0.02$ | 2.14 |
| SMOTE (Ratio%) | $0.75 \pm 1.40$ | $28.43 \pm 3.18$ | $0.12 \pm 0.37$ | $0.00 \pm 0.00$ | $3.18 \pm 1.83$ | $0.00 \pm 0.00$ | $0.00 \pm 0.00$ | 1.57 |
| TabCutMix (F1 Score%) | $92.67 \pm 0.22$ | $71.42 \pm 1.32$ | $82.47 \pm 0.35$ | $99.27 \pm 0.07$ | $60.33 \pm 0.25$ | $97.94 \pm 0.13$ | $99.94 \pm 0.01$ | 2.43 |
| TabCutMix (Ratio%) | $2.06 \pm 1.10$ | $39.47 \pm 6.70$ | $1.58 \pm 0.76$ | $0.61 \pm 0.03$ | $4.83 \pm 1.39$ | $0.00 \pm 0.00$ | $0.00 \pm 0.00$ | 3.14 |
| TabCutMixPlus (F1 Score%) | $92.63 \pm 0.20$ | $71.39 \pm 0.94$ | $82.28 \pm 0.39$ | $99.19 \pm 0.05$ | $60.39 \pm 0.17$ | $97.97 \pm 0.02$ | $99.95 \pm 0.03$ | 2.57 |
| TabCutMixPlus (Ratio%) | $0.36 \pm 0.27$ | $25.44 \pm 2.81$ | $0.70 \pm 0.39$ | $0.43 \pm 0.25$ | $3.88 \pm 0.19$ | $0.00 \pm 0.00$ | $0.00 \pm 0.00$ | 2.00 |

## D.5. Discussion on Memorization Ratio and DCR

The DCR metric measures the closest distance of each synthetic sample to the training and holdout sets, offering insights into potential privacy risks. Synthetic samples that closely resemble training data can indicate privacy concerns. However, DCR's dependence on the holdout set limits its robustness, as the results can vary with changes in the holdout set composition. This reliance underscores the need for alternative metrics that are less influenced by external data partitions.

To address this, we focus on the Memorization Ratio, which uses a distance ratio to detect overfitting by identifying synthetic samples that are disproportionately close to their nearest training neighbor compared to the second-closest. Unlike DCR, this metric is independent of the holdout set, directly assessing overfitting within the generative process. While the fixed threshold for the distance ratio is inspired by image generation literature, it provides a practical baseline for tabular data. We acknowledge that tailoring the threshold to account for tabular-specific features (e.g., categorical and numerical distributions) could improve its accuracy and plan to explore this in future work. Together, these metrics provide complementary insights into privacy risks and overfitting in generative models.

## D.6. Memorization Evaluation Metrics

In the context of generative modeling, memorization occurs when a model reproduces training samples too closely, rather than generating novel samples that reflect the underlying data distribution. While some degree of memorization might be acceptable or even desirable in certain scenarios (e.g., when high fidelity is required), excessive memorization can lead to overfitting, lack of diversity in generated samples, and potential privacy risks if sensitive data from the training set is replicated. Therefore, it is crucial to develop quantitative metrics to detect and measure memorization in generative models, especially for applications involving tabular data, where exact replication of training samples is particularly problematic.

To address this challenge, we propose the concept of the *memorization ratio* based on the relative distance ratio criterion. Let $x$ be a generated sample, and let $\mathcal{D}$ denote the training dataset. We define the distance ratio $r(x)$ of $x$ as:

$$r(x) = \frac{d(x, \mathrm{NN}_1(x, \mathcal{D}))}{d(x, \mathrm{NN}_2(x, \mathcal{D}))},$$

where $d(\cdot, \cdot)$ is a distance metric in the input sample space, $\mathrm{NN}_1(x, \mathcal{D})$ is the nearest neighbor of $x$ in $\mathcal{D}$, and $\mathrm{NN}_2(x, \mathcal{D})$ is the second-nearest neighbor of $x$ in $\mathcal{D}$. Intuitively, a small value of $r(x)$ indicates that the generated sample $x$ is nearly identical to a training sample, suggesting memorization. To formalize this notion, we follow the threshold of memorization in the image generation domain and consider a sample $x$ to be *memorized* if $r(x) < \frac{1}{3}$.

To quantify the extent of memorization across all generated samples, we compute the *memorization ratio*, defined as the proportion of generated samples that satisfy $r(x) < \frac{1}{3}$:

$$\text{Mem. Ratio} = \frac{1}{|\mathcal{G}|} \sum_{x \in \mathcal{G}} \mathbb{I}(r(x) < \frac{1}{3}),$$

where $\mathcal{G}$ is the set of generated samples and $\mathbb{I}(\cdot)$ is the indicator function.

While the memorization ratio provides a point estimate of memorization intensity at a fixed threshold $\frac{1}{3}$, it is also important to understand how the degree of memorization varies under different thresholds $\tau$. To this end, we propose the *Memorization Area Under Curve* (Mem-AUC). Mem-AUC is computed as:

$$\text{Mem-AUC} = \int_0^1 \text{Mem. Ratio}(\tau)\, d\tau,$$

where Memorization Ratio$(\tau)$ represents the proportion of generated samples for which $r(x) < \tau$ as a function of the threshold $\tau$. Mem-AUC captures the overall memorization behavior across a continuous range of thresholds. Higher Mem-AUC values indicate stronger memorization, while lower Mem-AUC values correspond to weaker memorization and better generalization.

## E. More Experimental Results

### E.1. Distance Ratio Distribution of TabCutMix

We analyze the distribution of the nearest-neighbor distance ratio, defined as $r = \frac{\mathrm{NN}_1(\boldsymbol{x}, \mathcal{D})}{\mathrm{NN}_2(\boldsymbol{x}, \mathcal{D})}$, to assess the severity of memorization. A more zero-concentrated ratio distribution indicates more severe memorization issue, as the generated sample $x$ is closer to a real sample in training set $\mathcal{D}$. Figure. 8 illustrates the distance ratio distribution for both the original TabSyn and TabSyn with TabCutMix, and we observe the following:

**Obs.1**: TabCutMix consistently shifts the distribution away from zero, indicating a reduction in memorization. For example, in the Magic dataset, TabCutMix reduces the memorization ratio from $80.01\%$ to $52.06\%$ by generating samples that are less tightly aligned with the real data in $\mathcal{D}$.

**Obs.2**: The distance ratio distributions for both TabSyn and TabSyn with TabCutMix exhibit a bipolar pattern, with a higher probability mass concentrated near 0 or 1, while the probability in the middle remains low. This indicates that more generated samples are either very close to real data points (suggesting memorization) or relatively far apart (suggesting diversity). In the Magic dataset, for instance, this bipolarization is prominent, with TabCutMix shifting a greater proportion of samples towards higher distance ratios, thus reducing memorization.

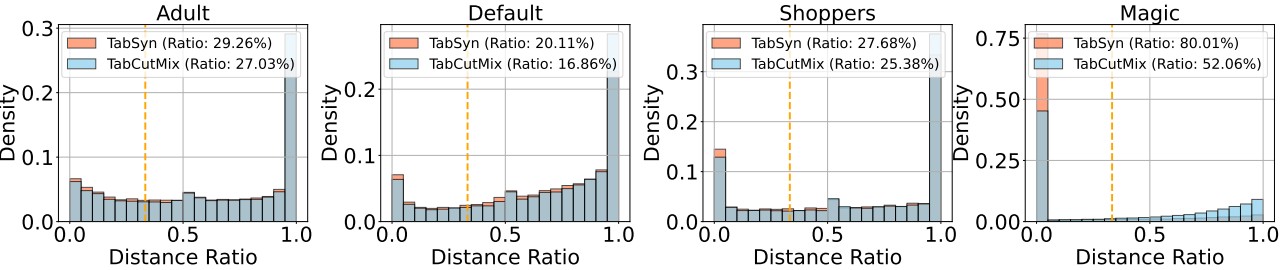

*Figure 8.* The nearest-neighbor distance ratio distributions of TabSyn with and without TabCutMix across different datasets.

## E.2. Data Distribution Comparison

Figure. 7 compares the distribution of real and synthetic data, with and without TabCutMix, for both numerical and categorical features across four datasets: Adult, Default, Shoppers, and Magic. We use one numerical feature and one categorical feature as examples from each dataset. We observe that

**Obs.1**: In the numerical feature distributions, TabCutMix generally synthesizes data, similar to w/o TabCutMix, aligned with the real data's distribution. For instance, in the Magic dataset, the Asym feature shows that the synthetic data generated by TabSyn has a good alignment with real data.

**Obs.2**: The categorical feature distributions show a similar improvement. In the Shoppers dataset, the proportion of values for the "VisitorType" feature generated by TabCutMix closely matches the real data, similar to the synthetic data generated without TabCutMix. This suggests that TabCutMix preserves the alignment between real and synthetic data for categorical features as well.

## E.3. Feature Correlation Matrix Comparison

Figure. 9 presents heatmaps of the pairwise column correlations between synthetic and real data. We compare the correlation matrices of synthetic data generated by TabSyn with TabCutMix against the real data. We observe that

**Obs.1**: TabCutMix preserves the quality of data generation in terms of correlation matrices, maintaining similar patterns to the synthetic data generated by TabSyn without introducing further errors. In datasets like Default and Shoppers, TabCutMix ensures that the synthetic data retains the essential correlation structure of the real data, without significant degradation in correlation matrix accuracy.

**Obs.2**: In the Magic dataset, while discrepancies between the synthetic and real data's correlation patterns persist, TabCutMix helps to maintain the existing data generation quality. Although it does not reduce the correlation matrix error, it ensures that the synthetic data continues to represent feature relationships similarly to TabSyn, preserving the overall structure.

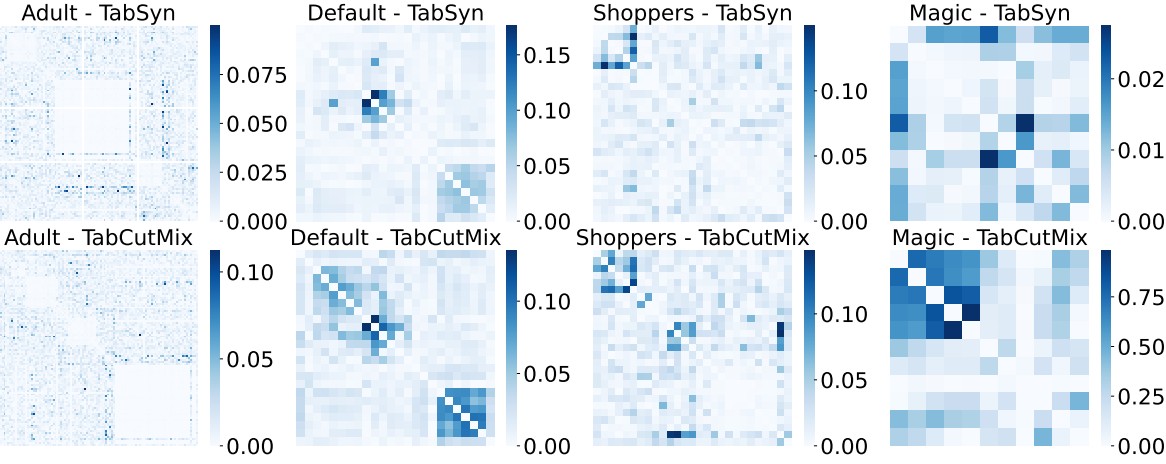

*Figure 9.* Heatmaps of the pair-wise column correlation of synthetic data v.s. the real data. The value represents the absolute divergence between the real and estimated correlations (the lighter, the better).

## E.4. More experimental Results on Shape Score

Figure 10 visualizes the shape scores for synthetic data generated by TabSyn and TabSyn combined with TabCutMix (w/ TCM) across multiple datasets (Adult, Default, Magic, and Shoppers). Shape scores reflect how closely the distribution of individual columns in synthetic data matches the real data. This figure compares these scores for different features to assess the fidelity of the generated data. We make the following observations:

**Obs. 1**: TabSyn and TabSyn+TabCutMix produce high-fidelity distributions across datasets. Across all datasets (Adult,

Default, Magic, Shoppers), both TabSyn and TabSyn+TabCutMix maintain high shape scores, suggesting that the generated samples from both methods capture the real data's feature distributions effectively. For instance, in the Adult and Default datasets, shape scores are consistently close to 1.0, indicating minimal divergence between synthetic and real data distributions.

**Obs. 2**: Low variance in shape scores across features. One notable observation across all datasets (Adult, Default, Magic, and Shoppers) is the consistently high shape scores across features, with minimal variance. For most features, the shape scores are very close to 1.0, indicating that both TabSyn and TabCutMix can replicate the real data distributions with high fidelity, regardless of feature type. The small variance in shape scores suggests that both methods generalize well across a wide range of features, from categorical to continuous, without significant degradation in performance for any particular feature.

The shape score comparison demonstrates that both TabSyn and TabCutMix generate synthetic data with high fidelity to the real data across multiple datasets.

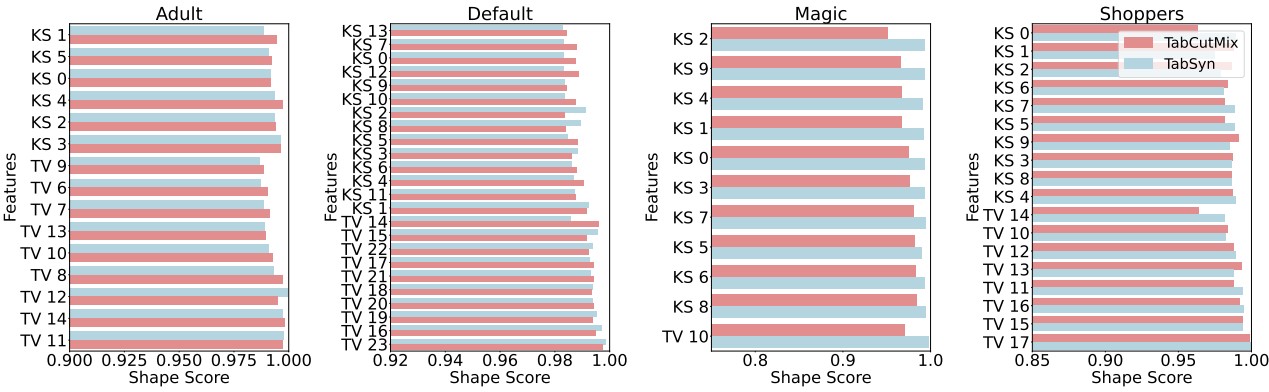

*Figure 10.* Shape score comparison for each feature in synthetic data generated by TabSyn and TabSyn+TabCutMix across multiple datasets.

## E.5. Case Study: Evaluating Augmented Data Quality with TabCutMix and TabCutMixPlus

To assess the quality of augmented data and identify potential issues with TabCutMix, we conducted a detailed case study using the Magic dataset, as summarized in Table 6. The table presents a comparison of real and generative samples produced by TabSyn with TabCutMix (denoted as TCM) and TabSyn with TabCutMixPlus (denoted as TCMP).

The results reveal that TabCutMix disrupts feature correlations, evident in unrealistic relationships like the Length being smaller than Width, which contradicts the inherent structure of the data. In contrast, TabCutMixPlus preserves feature coherence by clustering correlated features and swapping them within the same cluster. This approach ensures the generation of more realistic and consistent samples, demonstrating its effectiveness in maintaining data quality and utility compared to TabCutMix.

*Table 6.* The real and generative samples by TabSyn with TabCutMix and TabSyn with TabCutMixPlus in Magic dataset. TCM represents TabCutMix, TCMP represents TabCutMixPlus.

| Samples | Length | Width | Size | Conc | Conc1 | Asym | M3Long | M3Trans | Alpha | Dist | class |
|---|---|---|---|---|---|---|---|---|---|---|---|
| TabSyn+**TCM** | 24.72 | 32.12 | 3.35 | 0.15 | 0.09 | 150.69 | -40.91 | -21.80 | 5.11 | 205.42 | g |
| TabSyn+**TCMP** | 17.56 | 11.20 | 2.29 | 0.58 | 0.37 | 2.29 | 17.12 | 2.40 | 22.44 | 211.00 | g |

## E.6. Experimental Results on More Datasets

To broaden the evaluation of TabCutMix and TabCutMixPlus, we included 3 additional datasets Churn, Cardio, and Wilt. The results are summarized in Table 7, and the following observations are made:

**Obs. 1**: TabCutMix and TabCutMixPlus consistently reduce the memorization ratio across all datasets compared to various baselines. For instance, in the Churn dataset, the memorization ratio for TabSyn is reduced from 25.42% to 24.61% by

TabCutMix and to $24.79\%$ by TabCutMixPlus, representing an improvement of $3.21\%$ and $2.49\%$ compared to the vanilla method. This highlights the effectiveness of TabCutMix and TabCutMixPlus in mitigating memorization.

**Obs. 2**: For data quality, TabCutMixPlus achieves superior results in metrics such as MLE, $\alpha$-Precision, and $\beta$-Recall, indicating its ability to maintain data fidelity and enhance synthetic data utility. For example, in the Wilt dataset, TabCut-MixPlus achieves the highest $\alpha$-Precision ($99.10\%$) and $\beta$-Recall ($48.14\%$), reflecting its capability to preserve structural coherence while reducing memorization. In contrast, TabCutMix shows competitive but slightly lower performance (e.g., $98.73\%$ $\alpha$-Precision and $47.04\%$ $\beta$-Recall ), demonstrating that the clustering-based feature swaps in TabCutMixPlus offer additional advantages.

### E.7. More experiments on Baseline IJF

To compare the effectiveness of IJF with the proposed TabCutMix and TabCutMixPlus, we evaluate memorization and data generation quality using metrics such as memorization ratio, MLE, $\alpha$-Precision, $\beta$-Recall, shape score, and trend score across four datasets. The results are presented in Table 8, and we observe the following:

**Obs.1**: IJF demonstrates moderate success in reducing memorization ratios compared to the baseline generative models without augmentation. However, In terms of data quality metrics such as MLE, $\alpha$-Precision, and $\beta$-Recall, IJF performs weaker compared to TabCutMix and TabCutMixPlus in maintaining structural fidelity. For example, in the Adult dataset, the shape score for IJF ($96.55\%$) is slightly lower than that for TabCutMix ($97.26\%$) and TabCutMixPlus ($97.50\%$), suggesting that IJF may not fully capture the complex relationships between features that are preserved in the proposed methods.

**Obs.2**: IJF shows consistent performance across datasets but lags behind TabCutMix and TabCutMixPlus in scenarios where feature dependencies play a critical role. For instance, in the Shoppers dataset, TabCutMixPlus achieves the highest trend score ($97.77\%$), indicating its superiority in generating coherent and realistic data samples.

### E.8. Experimental Results on More Generative Models

The results in Table 9 demonstrate that **TabCutMixPlus (TCMP)** effectively reduces memorization across various generative models, including GAN-based models (e.g., CTGAN (Xu et al., 2019)) and VAE-based models (e.g., TVAE (Xu et al., 2019)). Specifically, for **CTGAN**, TCMP achieves a reduction in memorization ratio by $8.03\%$, $2.23\%$, $0.52\%$, and $6.70\%$ on the Default, Adult, Shoppers, and Magic datasets, respectively. Similarly, for TVAE, TCMP reduces memorization by $9.47\%$, $2.93\%$, $12.98\%$, and $8.98\%$ across the same datasets. These results indicate that **TabCutMixPlus** is not limited to diffusion models but is also broadly applicable to other types of generative models. By preserving feature correlations and addressing issues introduced by traditional TabCutMix, TCMP significantly mitigates memorization while maintaining or even improving key performance metrics such as MLE, shape score, and trend score.

### E.9. More Experiments on Mem-AUC

To verify the validity of using a fixed threshold of $\frac{1}{3}$ for measuring memorization, we computed both the Mem-AUC and the memorization ratio based on the $\frac{1}{3}$ threshold. We then evaluated the correlation between these two metrics. The results, as depicted in Figure 11, reveal a strong positive correlation across all scenarios. The high correlation between Mem-AUC and the memorization ratio supports the appropriateness of the $\frac{1}{3}$ threshold for practical applications. While Mem-AUC provides a more holistic evaluation by integrating memorization over the entire threshold range, the $\frac{1}{3}$ threshold remains a valid and reliable metric for assessing memorization, balancing simplicity and effectiveness in practice. Table 10 highlights the results for different augmentation techniques, including TabCutMixPlus, TabCutMix, Mixup, and SMOTE. These findings reinforce that the $\frac{1}{3}$ threshold remains a valid choice for simplicity and effectiveness in practice.

## F. Limitation Discussion

While this work makes significant contributions to augmenting tabular data with TabCutMix and its improved version, TabCutMixPlus, several limitations remain that warrant further exploration:

- Assumptions About Feature Independence. TabCutMix assumes that features can be swapped between samples independently without disrupting the data manifold. However, this assumption does not hold for datasets with strongly correlated features or complex interdependencies. For instance, in the Cardio dataset, the high feature correlation led to a relatively high OOD ratio ($4.83\%$), highlighting a limitation of the current method in preserving feature relationships

*Table 7.* The overview performance comparison for tabular diffusion models on more datasets. "TCM" represents our proposed **TabCutMix** and "TCMP" represents **TabCutMixPlus**. "Mem. Ratio" represents memorization ratio. "Improv" represents the improvement ratio on memorization.

| Dataset | Methods | Mem. Ratio (%)↓ | Improv. | MLE (%)↑ | α-Precision(%)↑ | β-Recall(%)↑ | Shape Score(%)↑ | Trend Score(%)↑ | C2ST(%)↑ | DCR(%) |
|---|---|---|---|---|---|---|---|---|---|---|
| Magic | STaSy | 77.52 ± 0.27 | - | 92.92 ± 0.30 | 91.18 ± 1.30 | 46.07 ± 1.61 | 88.12 ± 7.05 | 90.27 ± 6.53 | 75.02 ± 4.03 | 52.57 ± 0.92 |
| | STaSy+**Mixup** | 78.11 ± 0.18 | −0.77%↓ | 93.03 ± 0.16 | 91.03 ± 3.57 | 50.19 ± 0.84 | 94.30 ± 1.91 | 96.67 ± 1.10 | 79.72 ± 6.80 | 50.27 ± 1.42 |
| | STaSy+**SMOTE** | 76.88 ± 0.43 | 0.83%↓ | 92.95 ± 1.59 | 67.32 ± 1.04 | 52.39 ± 2.18 | 88.78 ± 0.91 | 89.78 ± 1.45 | 53.08 ± 3.70 | 51.24 ± 0.74 |
| | STaSy+**TCM** | 75.12 ± 0.29 | 3.10%↓ | 91.49 ± 0.63 | 92.50 ± 3.01 | 35.24 ± 1.48 | 89.62 ± 5.33 | 89.96 ± 6.44 | 75.70 ± 5.53 | 49.85 ± 0.21 |
| | STaSy+**TCMP** | 76.70 ± 0.38 | 1.06%↓ | 92.77 ± 0.20 | 97.27 ± 1.30 | 40.11 ± 1.65 | 95.37 ± 1.51 | 96.34 ± 0.42 | 76.63 ± 6.85 | 48.41 ± 0.28 |
| | TabDDPM | 77.62 ± 2.11 | - | 92.78 ± 0.23 | 98.41 ± 0.37 | 46.67 ± 1.18 | 99.07 ± 0.06 | 98.58 ± 0.51 | 99.05 ± 0.70 | 50.47 ± 0.42 |
| | TabDDPM+**Mixup** | 78.37 ± 0.81 | −0.97%↓ | 92.08 ± 0.58 | 92.01 ± 1.24 | 45.45 ± 1.38 | 96.22 ± 0.53 | 97.26 ± 1.69 | 98.33 ± 2.34 | 50.92 ± 0.20 |
| | TabDDPM+**SMOTE** | 72.31 ± 1.56 | 6.84%↓ | 91.68 ± 0.52 | 66.45 ± 3.04 | 45.35 ± 2.70 | 89.30 ± 0.77 | 88.04 ± 1.54 | 54.27 ± 0.56 | 50.83 ± 0.71 |
| | TabDDPM+**TCM** | 72.99 ± 0.22 | 5.96%↓ | 91.69 ± 0.86 | 97.92 ± 0.38 | 32.51 ± 0.70 | 98.97 ± 0.08 | 99.19 ± 0.11 | 97.62 ± 2.44 | 49.73 ± 0.35 |
| | TabDDPM+**TCMP** | 76.22 ± 0.39 | 1.81%↓ | 91.50 ± 0.22 | 96.50 ± 4.02 | 36.52 ± 2.43 | 98.08 ± 1.32 | 95.17 ± 3.05 | 95.29 ± 6.22 | 49.88 ± 0.74 |
| | TabSyn | 80.02 ± 0.39 | - | 93.18 ± 0.31 | 99.10 ± 0.68 | 48.28 ± 0.41 | 99.00 ± 0.28 | 99.15 ± 0.08 | 99.75 ± 0.29 | 50.48 ± 0.16 |
| | TabSyn+**Mixup** | 78.88 ± 0.78 | 1.42%↓ | 92.63 ± 0.45 | 91.68 ± 0.20 | 48.50 ± 0.17 | 96.70 ± 0.10 | 98.41 ± 0.34 | 99.68 ± 0.43 | 51.01 ± 0.51 |
| | TabSyn+**SMOTE** | 72.14 ± 1.27 | 9.85%↓ | 92.74 ± 0.12 | 63.32 ± 0.79 | 48.73 ± 2.69 | 89.36 ± 0.86 | 89.02 ± 0.89 | 56.26 ± 2.53 | 50.29 ± 0.53 |
| | TabSyn+**TCM** | 52.06 ± 7.12 | 34.94%↓ | 91.77 ± 0.12 | 96.83 ± 0.40 | 30.79 ± 2.92 | 97.83 ± 0.65 | 98.09 ± 0.17 | 93.55 ± 1.49 | 51.76 ± 0.49 |
| | TabSyn+**TCMP** | 76.46 ± 0.36 | 4.44%↓ | 91.91 ± 0.42 | 98.03 ± 1.76 | 39.54 ± 1.54 | 98.87 ± 0.57 | 97.26 ± 0.27 | 97.58 ± 3.36 | 51.32 ± 0.63 |
| Churn | STaSy | 27.01 ± 0.30 | - | 84.80 ± 2.24 | 92.39 ± 2.97 | 37.42 ± 8.07 | 87.17 ± 6.64 | 86.95 ± 5.99 | 48.42 ± 10.89 | 50.70 ± 2.00 |
| | STaSy+**Mixup** | 24.86 ± 3.47 | 7.97%↓ | 85.08 ± 2.46 | 89.09 ± 1.40 | 46.08 ± 2.81 | 87.44 ± 2.82 | 87.95 ± 0.58 | 47.19 ± 7.58 | 51.79 ± 0.48 |
| | STaSy+**SMOTE** | 22.36 ± 0.87 | 17.20%↓ | 83.73 ± 1.37 | 82.44 ± 1.78 | 35.85 ± 4.59 | 84.51 ± 3.71 | 86.23 ± 0.43 | 40.76 ± 4.83 | 51.90 ± 0.58 |
| | STaSy+**IJF** | 22.96 ± 2.42 | 14.98%↓ | 82.62 ± 0.77 | 82.90 ± 1.10 | 41.34 ± 2.21 | 87.23 ± 0.63 | 70.33 ± 2.38 | 48.53 ± 1.61 | 47.25 ± 0.43 |
| | STaSy+**TCM** | 22.86 ± 2.32 | 15.36%↓ | 84.01 ± 2.62 | 96.22 ± 2.87 | 43.16 ± 1.19 | 91.03 ± 1.60 | 90.30 ± 1.57 | 49.73 ± 4.26 | 52.26 ± 1.29 |
| | STaSy+**TCMP** | 24.12 ± 1.05 | 10.69%↓ | 85.36 ± 2.16 | 94.92 ± 3.45 | 43.61 ± 2.21 | 91.10± 1.20 | 90.22 ± 0.76 | 50.68 ± 0.79 | 50.10 ± 1.80 |
| | TabDDPM | 25.43 ± 1.00 | - | 86.31 ± 2.57 | 99.10 ± 0.36 | 51.03 ± 0.90 | 98.84 ± 0.30 | 98.06 ± 0.37 | 98.71 ± 1.40 | 49.23± 2.27 |
| | TabDDPM+**Mixup** | 25.00 ± 0.68 | 1.69%↓ | 85.99 ± 1.49 | 95.04 ± 2.25 | 49.52 ± 1.49 | 95.98 ± 1.37 | 93.84± 2.43 | 88.77 ± 3.39 | 48.73 ± 0.82 |
| | TabDDPM+**SMOTE** | 24.57 ± 0.61 | 3.41%↓ | 84.57 ± 2.14 | 86.46 ± 3.22 | 46.25 ± 1.18 | 94.53 ± 0.90 | 92.02 ± 2.19 | 80.12 ± 3.96 | 50.42 ± 0.23 |
| | TabDDPM+**IJF** | 24.96 ± 0.53 | 1.86%↓ | 85.36 ± 1.78 | 89.12 ± 9.43 | 43.81 ± 4.20 | 93.40 ± 6.26 | 73.58 ± 2.61 | 89.12 ± 8.12 | 50.86 ± 0.84 |
| | TabDDPM+**TCM** | 24.42 ± 0.71 | 4.00%↓ | 86.39 ± 1.82 | 98.51 ± 0.87 | 50.41 ± 0.75 | 98.55 ± 0.52 | 98.18 ± 0.81 | 96.62 ± 3.13 | 52.98 ± 0.53 |
| | TabDDPM+**TCMP** | 24.66 ± 0.32 | 3.03%↓ | 86.55 ± 2.96 | 98.13 ± 1.21 | 51.82 ± 2.19 | 98.15 ± 0.33 | 97.78 ± 0.27 | 96.62 ± 2.20 | 50.37 ± 2.43 |
| | TabSyn | 25.42 ± 0.21 | - | 86.04 ± 2.38 | 99.31 ± 0.31 | 50.45 ± 1.06 | 99.14 ± 0.13 | 98.15 ± 0.19 | 99.89 ± 0.06 | 50.80 ± 0.61 |
| | TabSyn+**Mixup** | 24.74 ± 0.51 | 2.70%↓ | 85.84 ± 1.56 | 98.37 ± 0.16 | 49.43 ± 0.74 | 98.02 ± 0.65 | 97.10 ± 0.73 | 95.94 ± 5.81 | 50.43 ± 0.25 |
| | TabSyn+**SMOTE** | 24.87 ± 0.46 | 2.19%↓ | 85.60 ± 1.97 | 98.63 ± 0.50 | 44.68 ± 0.40 | 98.25 ± 0.34 | 75.00 ± 1.50 | 99.08 ± 0.79 | 50.73 ± 1.42 |
| | TabSyn+**IJF** | 24.53 ± 0.38 | 3.50%↓ | 83.12 ± 1.76 | 88.51 ± 0.40 | 46.62 ± 1.02 | 94.98 ± 0.54 | 93.31 ± 0.34 | 83.22 ± 2.14 | 52.28 ± 0.45 |
| | TabSyn+**TCM** | 24.61 ± 0.17 | 3.21%↓ | 85.60 ± 2.41 | 99.12 ± 0.46 | 49.60 ± 0.38 | 99.14 ± 0.19 | 98.25 ± 0.28 | 99.70 ± 0.33 | 52.15 ± 0.77 |
| | TabSyn+**TCMP** | 24.79 ± 0.15 | 2.49%↓ | 86.28 ± 2.15 | 99.10 ± 0.28 | 49.62± 0.64 | 98.99 ± 0.50 | 98.70 ± 0.32 | 99.49 ± 0.32 | 49.30 ± 1.90 |
| Cardio | STaSy | 23.94 ± 0.12 | - | 79.89 ± 0.74 | 93.72 ± 3.19 | 46.46 ± 0.87 | 96.17 ± 1.24 | 96.37 ± 0.80 | 85.00 ± 4.66 | 50.10 ± 0.71 |
| | STaSy+**Mixup** | 23.84 ± 0.15 | 0.42%↓ | 79.30 ± 0.28 | 94.86 ± 3.17 | 46.32 ± 0.47 | 95.14 ± 0.79 | 95.90 ± 0.15 | 82.90 ± 3.11 | 49.79 ± 0.53 |
| | STaSy+**SMOTE** | 22.77 ± 0.27 | 4.87%↓ | 78.81 ± 0.18 | 90.57 ± 3.00 | 45.11 ± 1.24 | 95.37 ± 0.48 | 95.16 ± 1.34 | 76.82 ± 5.21 | 50.48 ± 0.24 |
| | STaSy+**IJF** | 22.51 ± 1.12 | 5.99%↓ | 79.63 ± 0.91 | 95.48 ± 1.24 | 40.96 ± 0.19 | 96.24 ± 0.43 | 92.42 ± 0.47 | 85.98 ± 4.76 | 50.54 ± 1.51 |
| | STaSy+**TCM** | 22.50 ± 0.35 | 6.00%↓ | 79.71 ± 0.46 | 95.11 ± 3.87 | 45.92 ± 1.76 | 95.79 ± 2.18 | 95.95 ± 1.74 | 85.34 ± 2.49 | 52.24 ± 3.03 |
| | STaSy+**TCMP** | 22.81 ± 0.48 | 4.73%↓ | 79.96 ± 0.33 | 94.93 ± 2.82 | 46.09 ± 2.68 | 96.37± 1.59 | 96.07 ± 1.06 | 85.99 ± 2.21 | 50.43 ± 0.74 |
| | TabDDPM | 24.63 ± 0.18 | - | 80.24 ± 0.78 | 99.14 ± 0.15 | 49.11 ± 0.17 | 99.61 ± 0.03 | 98.95 ± 0.30 | 99.43 ± 0.55 | 49.91 ± 0.36 |
| | TabDDPM+**Mixup** | 24.00 ± 0.36 | 2.57%↓ | 79.62 ± 0.27 | 99.22 ± 0.45 | 48.17 ± 0.17 | 97.80 ± 0.72 | 97.38 ± 0.85 | 95.01 ± 2.25 | 50.15 ± 0.54 |
| | TabDDPM+**SMOTE** | 23.10 ± 0.69 | 6.21%↓ | 79.47 ± 0.45 | 96.35 ± 2.38 | 47.44 ± 1.17 | 96.58 ± 0.75 | 94.39 ± 1.28 | 85.43 ± 1.69 | 50.73 ± 0.19 |
| | TabDDPM+**IJF** | 22.00 ± 0.83 | 10.68%↓ | 79.37 ± 0.79 | 98.12 ± 0.27 | 42.58 ± 0.33 | 97.87 ± 0.13 | 94.49 ± 0.32 | 99.46 ± 0.16 | 50.10 ± 0.48 |
| | TabDDPM+**TCM** | 23.05 ± 0.38 | 6.40%↓ | 79.71 ± 0.58 | 97.82 ± 2.05 | 48.37 ± 1.11 | 98.66 ± 1.35 | 95.86 ± 3.43 | 96.31 ± 0.42 | 49.24 ± 1.58 |
| | TabDDPM+**TCMP** | 23.54 ± 0.34 | 4.43%↓ | 79.82 ± 0.27 | 98.71 ± 0.49 | 48.87 ± 0.34 | 98.88 ± 0.62 | 98.67 ± 0.20 | 96.31 ± 0.42 | 49.34 ± 0.38 |
| | TabSyn | 25.31 ± 0.45 | - | 80.04 ± 0.79 | 95.70 ± 2.65 | 49.63 ± 0.69 | 97.43 ± 0.76 | 96.63 ± 1.67 | 91.46 ± 1.99 | 50.34 ± 0.79 |
| | TabSyn+**Mixup** | 24.46 ± 0.43 | 3.33%↓ | 79.85 ± 0.30 | 98.48 ± 0.78 | 48.15 ± 0.56 | 98.32 ± 0.53 | 96.10 ± 2.38 | 96.64 ± 3.19 | 50.69 ± 0.34 |
| | TabSyn+**SMOTE** | 23.85 ± 0.68 | 5.77%↓ | 79.76 ± 0.49 | 97.54 ± 0.82 | 47.47 ± 0.44 | 97.06 ± 0.71 | 95.00 ± 3.33 | 86.30 ± 3.50 | 48.73 ± 0.36 |
| | TabSyn+**IJF** | 22.53 ± 0.80 | 10.97%↓ | 79.43 ± 0.15 | 98.09 ± 0.17 | 42.83 ± 0.32 | 97.96 ± 0.17 | 94.62 ± 0.60 | 99.49 ± 0.19 | 50.48 ± 0.36 |
| | TabSyn+**TCM** | 22.97 ± 0.17 | 9.23%↓ | 79.92 ± 0.44 | 98.59 ± 0.98 | 48.62 ± 0.76 | 98.90 ± 0.63 | 97.93 ± 0.99 | 95.97 ± 3.13 | 50.27 ± 1.22 |
| | TabSyn+**TCMP** | 23.90 ± 0.42 | 5.55%↓ | 79.79 ± 0.60 | 98.32 ± 0.36 | 48.60 ± 0.90 | 98.55 ± 0.14 | 98.12 ± 0.99 | 95.78 ± 0.60 | 49.95 ± 0.27 |
| Witt | STaSy | 98.42 ± 0.24 | - | 98.74 ± 1.15 | 86.68 ± 5.50 | 42.20 ± 1.00 | 82.39 ± 8.05 | 91.16 ± 5.11 | 36.64 ± 6.77 | 52.27 ± 5.18 |
| | STaSy+**Mixup** | 97.61 ± 0.81 | 0.77%↓ | 99.05 ± 0.84 | 91.78 ± 8.33 | 43.48 ± 3.40 | 85.41 ± 4.54 | 88.76 ± 1.65 | 47.88 ± 6.39 | 45.45 ± 8.90 |
| | STaSy+**SMOTE** | 97.31 ± 0.82 | 1.13%↓ | 98.88 ± 0.71 | 76.07 ± 2.06 | 36.07 ± 2.25 | 80.96 ± 2.64 | 85.34 ± 5.12 | 35.63 ± 3.54 | 50.49 ± 0.33 |
| | STaSy+**IJF** | 96.53 ± 0.31 | 1.92%↓ | 96.86 ± 1.75 | 79.48 ± 6.34 | 35.22 ± 7.30 | 79.20 ± 4.54 | 79.41 ± 4.02 | 37.17 ± 10.11 | 54.27 ± 1.13 |
| | STaSy+**TCM** | 92.47 ± 5.97 | 6.05%↓ | 98.80 ± 0.64 | 91.10 ± 9.72 | 42.21 ± 8.70 | 87.34 ± 12.46 | 91.68 ± 7.60 | 47.49 ± 14.75 | 51.65 ± 2.80 |
| | STaSy+**TCMP** | 97.60 ± 0.84 | 0.84%↓ | 99.33 ± 0.31 | 90.66 ± 7.37 | 42.22 ± 9.77 | 85.32 ± 8.57 | 90.94 ± 7.43 | 49.98 ± 4.59 | 48.19 ± 3.51 |
| | TabDDPM | 98.48 ± 0.35 | - | 99.32 ± 0.58 | 98.63 ± 0.73 | 50.53 ± 0.47 | 98.58 ± 1.51 | 98.48 ± 0.35 | 98.63 ± 1.68 | 52.47 ± 0.54 |
| | TabDDPM+**Mixup** | 98.16 ± 0.24 | 0.32%↓ | 99.34 ± 0.44 | 96.29 ± 0.49 | 52.13 ± 1.11 | 97.34 ± 0.99 | 92.24 ± 3.21 | 96.05 ± 1.97 | 50.84 ± 0.26 |
| | TabDDPM+**SMOTE** | 96.78 ± 0.41 | 1.72%↓ | 99.22 ± 0.54 | 79.49 ± 0.78 | 43.76 ± 1.37 | 91.09 ± 0.50 | 88.04 ± 4.73 | 76.66 ± 2.61 | 50.29 ± 0.28 |
| | TabDDPM+**IJF** | 94.16 ± 0.18 | 4.39%↓ | 98.90 ± 0.66 | 96.51 ± 0.63 | 43.95 ± 1.12 | 96.69 ± 0.26 | 86.63 ± 0.61 | 98.57 ± 0.87 | 52.46 ± 3.66 |
| | TabDDPM+**TCM** | 97.17 ± 0.12 | 1.33%↓ | 99.22 ± 0.38 | 97.93 ± 1.01 | 48.47 ± 1.17 | 97.31 ± 1.28 | 95.71 ± 2.49 | 96.92 ± 1.72 | 48.75 ± 2.18 |
| | TabDDPM+**TCMP** | 96.75 ± 0.67 | 1.76%↓ | 99.52 ± 0.37 | 98.55 ± 0.14 | 49.36 ± 0.99 | 97.12 ± 0.84 | 96.81 ± 0.63 | 96.76 ± 3.44 | 45.52 ± 1.35 |
| | TabSyn | 97.67 ± 0.39 | - | 99.85 ± 0.07 | 98.83 ± 0.28 | 47.96 ± 0.64 | 98.73 ± 0.12 | 98.62 ± 0.19 | 99.91 ± 0.07 | 51.71 ± 2.94 |
| | TabSyn+**Mixup** | 97.62 ± 0.22 | 0.05%↓ | 99.44 ± 0.34 | 97.03 ± 0.13 | 48.89 ± 1.61 | 98.51 ± 0.08 | 93.38 ± 4.39 | 98.85 ± 0.14 | 50.68 ± 0.42 |
| | TabSyn+**SMOTE** | 94.84 ± 0.60 | 2.89%↓ | 99.13 ± 0.84 | 77.86 ± 1.27 | 41.96 ± 0.26 | 90.57 ± 0.61 | 87.85 ± 3.33 | 77.73 ± 1.02 | 47.10 ± 2.18 |
| | TabSyn+**IJF** | 93.62 ± 0.52 | 4.14%↓ | 99.06 ± 0.65 | 96.48 ± 0.22 | 42.12 ± 0.69 | 97.28 ± 0.64 | 85.05 ± 0.22 | 99.64 ± 0.35 | 51.66 ± 0.89 |
| | TabSyn+**TCM** | 95.95 ± 0.19 | 1.76%↓ | 99.45 ± 0.29 | 98.73 ± 0.51 | 47.04 ± 0.25 | 98.64 ± 0.06 | 98.11 ± 0.30 | 99.73 ± 0.24 | 49.44 ± 4.08 |
| | TabSyn+**TCMP** | 96.79 ± 0.23 | 0.90%↓ | 99.70 ± 0.14 | 99.10 ± 0.23 | 48.14 ± 0.41 | 98.47 ± 0.29 | 98.74 ± 0.07 | 99.06 ± 0.78 | 49.72 ± 1.02 |

*Table 8.* The overview performance comparison for tabular diffusion models on IJF and our proposed methods. "TCM" represents our proposed **TabCutMix** and "TCMP" represents **TabCutMixPlus**. "Mem. Ratio" represents the memorization ratio. "Improv" represents the improvement ratio on memorization.

| | Methods | Mem. Ratio (%)↓ | Improv. | MLE (%)↑ | $\alpha$-Precision(%)↑ | $\beta$-Recall(%)↑ | Shape Score(%)↑ | Trend Score(%)↑ | C2ST(%)↑ | DCR(%) |
|---|---|---|---|---|---|---|---|---|---|---|
| Default | STaSy | $17.57 \pm 0.53$ | - | $76.48 \pm 1.18$ | $87.78 \pm 5.20$ | $35.94 \pm 5.48$ | $90.27 \pm 2.43$ | $89.58 \pm 1.35$ | $67.68 \pm 6.89$ | $50.30 \pm 0.36$ |
| | STaSy+**IJF** | $14.59 \pm 1.77$ | **16.94%** ↓ | $75.15 \pm 0.96$ | $86.59 \pm 3.71$ | $31.56 \pm 0.16$ | $89.19 \pm 1.12$ | $31.42 \pm 0.72$ | $49.28 \pm 1.46$ | $51.30 \pm 2.78$ |
| | STaSy+**TCM** | $14.51 \pm 0.46$ | **17.44%** ↓ | $75.33 \pm 1.32$ | $86.04 \pm 11.55$ | $32.13 \pm 5.07$ | $90.30 \pm 3.88$ | $89.85 \pm 3.16$ | $49.51 \pm 6.33$ | $50.39 \pm 0.99$ |
| | STaSy+**TCMP** | $15.53 \pm 2.00$ | **11.59%** ↓ | $76.30 \pm 0.57$ | $90.83 \pm 4.51$ | $32.81 \pm 1.37$ | $91.49 \pm 0.77$ | $92.08 \pm 2.04$ | $50.43 \pm 2.00$ | $50.70 \pm 1.94$ |
| | TabDDPM | $19.33 \pm 0.45$ | - | $76.79 \pm 0.69$ | $98.15 \pm 1.45$ | $44.41 \pm 0.70$ | $97.58 \pm 0.95$ | $94.46 \pm 0.68$ | $91.85 \pm 6.04$ | $49.12 \pm 0.94$ |
| | TabDDPM+**IJF** | $14.02 \pm 1.12$ | **27.47%** ↓ | $76.19 \pm 0.87$ | $93.82 \pm 0.68$ | $38.59 \pm 0.49$ | $96.36 \pm 0.53$ | $28.49 \pm 1.27$ | $95.91 \pm 1.77$ | $49.86 \pm 1.22$ |
| | TabDDPM+**TCM** | $16.76 \pm 0.47$ | **13.26%** ↓ | $76.47 \pm 0.60$ | $97.30 \pm 0.46$ | $38.72 \pm 2.78$ | $97.27 \pm 1.74$ | $93.27 \pm 2.52$ | $94.72 \pm 3.87$ | $50.23 \pm 0.53$ |
| | TabDDPM+**TCMP** | $18.00 \pm 0.24$ | **6.88%** ↓ | $76.92 \pm 0.17$ | $98.26 \pm 0.25$ | $41.92 \pm 0.52$ | $97.37 \pm 0.09$ | $91.42 \pm 1.15$ | $95.64 \pm 0.49$ | $49.75 \pm 0.32$ |
| | TabSyn | $20.11 \pm 0.03$ | - | $77.00 \pm 0.33$ | $98.66 \pm 0.13$ | $46.76 \pm 0.50$ | $98.96 \pm 0.11$ | $96.82 \pm 1.71$ | $98.27 \pm 1.14$ | $51.09 \pm 0.32$ |
| | TabSyn+**IJF** | $15.82 \pm 0.33$ | **21.33%** ↓ | $76.53 \pm 0.56$ | $92.66 \pm 0.39$ | $39.11 \pm 0.35$ | $96.51 \pm 0.30$ | $31.84 \pm 0.31$ | $97.64 \pm 0.80$ | $49.58 \pm 0.67$ |
| | TabSyn+**TCM** | $16.86 \pm 1.36$ | **16.16%** ↓ | $76.84 \pm 0.34$ | $96.16 \pm 1.24$ | $40.69 \pm 2.46$ | $98.02 \pm 1.62$ | $96.51 \pm 1.42$ | $97.65 \pm 0.65$ | $51.16 \pm 1.82$ |
| | TabSyn+**TCMP** | $17.60 \pm 0.28$ | **12.48%** ↓ | $77.17 \pm 0.51$ | $97.61 \pm 0.27$ | $44.46 \pm 0.60$ | $99.03 \pm 0.08$ | $96.30 \pm 1.48$ | $98.16 \pm 0.65$ | $51.20 \pm 0.90$ |
| Adult | STaSy | $26.02 \pm 0.89$ | - | $90.54 \pm 0.17$ | $85.79 \pm 7.85$ | $34.35 \pm 2.46$ | $89.14 \pm 2.29$ | $86.00 \pm 2.97$ | $51.89 \pm 14.87$ | $50.46 \pm 0.39$ |
| | STaSy+**IJF** | $20.80 \pm 2.03$ | **20.07%** ↓ | $90.55 \pm 0.18$ | $81.69 \pm 10.52$ | $28.41 \pm 5.36$ | $88.19 \pm 3.23$ | $58.59 \pm 1.91$ | $45.93 \pm 10.41$ | $50.46 \pm 0.32$ |
| | STaSy+**TCM** | $20.89 \pm 1.33$ | **19.71%** ↓ | $90.45 \pm 0.30$ | $85.39 \pm 1.61$ | $31.24 \pm 0.97$ | $88.33 \pm 3.63$ | $85.39 \pm 4.03$ | $45.49 \pm 4.78$ | $50.92 \pm 0.39$ |
| | STaSy+**TCMP** | $21.45 \pm 2.60$ | **17.59%** ↓ | $90.72 \pm 0.06$ | $86.71 \pm 4.12$ | $32.63 \pm 1.81$ | $89.62 \pm 1.55$ | $86.05 \pm 2.44$ | $49.12 \pm 9.95$ | $50.75 \pm 0.59$ |
| | TabDDPM | $31.01 \pm 0.18$ | - | $91.09 \pm 0.07$ | $93.58 \pm 1.99$ | $51.52 \pm 2.29$ | $98.84 \pm 0.03$ | $97.78 \pm 0.07$ | $94.63 \pm 1.19$ | $51.56 \pm 0.34$ |
| | TabDDPM+**IJF** | $24.98 \pm 0.41$ | **19.45%** ↓ | $89.96 \pm 0.52$ | $95.32 \pm 0.18$ | $42.57 \pm 0.30$ | $97.45 \pm 0.66$ | $62.80 \pm 0.97$ | $95.68 \pm 0.49$ | $50.47 \pm 0.27$ |
| | TabDDPM+**TCM** | $27.55 \pm 0.19$ | **11.16%** ↓ | $91.15 \pm 0.06$ | $94.97 \pm 0.06$ | $47.43 \pm 1.46$ | $98.65 \pm 0.03$ | $97.75 \pm 0.07$ | $85.61 \pm 16.03$ | $50.99 \pm 0.65$ |
| | TabDDPM+**TCMP** | $26.10 \pm 2.11$ | **15.83%** ↓ | $90.54 \pm 0.17$ | $92.26 \pm 6.97$ | $43.49 \pm 3.74$ | $95.10 \pm 4.27$ | $91.50 \pm 6.53$ | $84.76 \pm 10.12$ | $50.68 \pm 0.89$ |
| | TabSyn | $29.26 \pm 0.23$ | - | $91.13 \pm 0.09$ | $99.31 \pm 0.39$ | $48.00 \pm 0.22$ | $99.33 \pm 0.09$ | $98.19 \pm 0.50$ | $98.68 \pm 0.41$ | $50.42 \pm 0.27$ |
| | TabSyn+**IJF** | $24.71 \pm 0.80$ | **15.53%** ↓ | $90.82 \pm 0.13$ | $98.94 \pm 0.38$ | $41.61 \pm 0.57$ | $97.40 \pm 0.54$ | $63.93 \pm 1.19$ | $99.55 \pm 0.27$ | $50.57 \pm 0.43$ |
| | TabSyn+**TCM** | $27.03 \pm 0.22$ | **7.60%** ↓ | $91.09 \pm 0.17$ | $99.04 \pm 0.42$ | $44.95 \pm 0.42$ | $99.40 \pm 0.07$ | $98.51 \pm 0.08$ | $89.18 \pm 1.94$ | $50.67 \pm 0.11$ |
| | TabSyn+**TCMP** | $25.99 \pm 0.52$ | **11.17%** ↓ | $90.96 \pm 0.16$ | $98.43 \pm 1.04$ | $43.23 \pm 2.96$ | $98.38 \pm 0.91$ | $96.53 \pm 1.47$ | $93.39 \pm 6.01$ | $50.30 \pm 0.78$ |
| Shoppers | STaSy | $25.51 \pm 0.32$ | - | $91.26 \pm 0.23$ | $88.02 \pm 3.54$ | $34.58 \pm 1.84$ | $88.18 \pm 0.29$ | $89.10 \pm 0.53$ | $47.85 \pm 8.48$ | $51.68 \pm 0.56$ |
| | STaSy+**IJF** | $23.71 \pm 0.39$ | **7.06%** ↓ | $90.28 \pm 0.95$ | $85.79 \pm 6.83$ | $34.04 \pm 7.18$ | $87.15 \pm 4.48$ | $51.55 \pm 1.20$ | $50.70 \pm 12.74$ | $50.29 \pm 0.20$ |
| | STaSy+**TCM** | $22.78 \pm 0.69$ | **10.71%** ↓ | $90.56 \pm 0.44$ | $86.66 \pm 4.18$ | $34.08 \pm 1.46$ | $87.16 \pm 3.78$ | $86.56 \pm 4.26$ | $50.08 \pm 6.30$ | $50.61 \pm 0.41$ |
| | STaSy+**TCMP** | $22.19 \pm 1.21$ | **13.03%** ↓ | $91.37 \pm 0.65$ | $85.82 \pm 2.66$ | $34.11 \pm 2.08$ | $87.38 \pm 2.30$ | $88.61 \pm 1.64$ | $52.42 \pm 2.65$ | $51.19 \pm 0.95$ |
| | TabDDPM | $31.37 \pm 0.31$ | - | $92.17 \pm 0.32$ | $93.16 \pm 1.58$ | $52.57 \pm 1.30$ | $97.08 \pm 0.46$ | $92.92 \pm 3.27$ | $86.74 \pm 0.63$ | $51.36 \pm 0.63$ |
| | TabDDPM+**IJF** | $26.45 \pm 0.61$ | **15.67%** ↓ | $91.29 \pm 0.43$ | $87.90 \pm 0.43$ | $46.28 \pm 0.89$ | $95.18 \pm 0.60$ | $58.69 \pm 4.57$ | $85.11 \pm 0.54$ | $50.44 \pm 2.24$ |
| | TabDDPM+**TCM** | $25.56 \pm 1.17$ | **18.51%** ↓ | $92.17 \pm 0.26$ | $94.41 \pm 1.49$ | $50.05 \pm 1.59$ | $97.18 \pm 0.34$ | $93.95 \pm 0.51$ | $86.96 \pm 0.50$ | $47.52 \pm 1.81$ |
| | TabDDPM+**TCMP** | $28.51 \pm 0.35$ | **9.12%** ↓ | $92.09 \pm 0.99$ | $93.43 \pm 1.65$ | $52.30 \pm 0.73$ | $97.31 \pm 0.22$ | $94.79 \pm 0.30$ | $87.02 \pm 2.04$ | $50.83 \pm 0.59$ |
| | TabSyn | $27.68 \pm 0.10$ | - | $91.76 \pm 0.66$ | $99.20 \pm 0.29$ | $47.79 \pm 0.77$ | $98.54 \pm 0.19$ | $97.83 \pm 0.10$ | $95.44 \pm 0.39$ | $52.50 \pm 0.44$ |
| | TabSyn+**IJF** | $25.54 \pm 0.28$ | **7.73%** ↓ | $91.41 \pm 1.01$ | $98.46 \pm 0.93$ | $43.72 \pm 4.30$ | $96.22 \pm 1.85$ | $96.39 \pm 1.80$ | $94.05 \pm 4.58$ | $51.35 \pm 1.61$ |
| | TabSyn+**TCM** | $25.38 \pm 0.18$ | **8.30%** ↓ | $91.43 \pm 0.26$ | $99.11 \pm 0.28$ | $45.98 \pm 0.90$ | $98.56 \pm 0.10$ | $97.85 \pm 0.06$ | $97.28 \pm 2.41$ | $49.92 \pm 1.59$ |
| | TabSyn+**TCMP** | $25.93 \pm 0.23$ | **6.33%** ↓ | $91.75 \pm 0.47$ | $99.24 \pm 0.55$ | $46.48 \pm 0.77$ | $98.60 \pm 0.14$ | $97.77 \pm 0.09$ | $97.40 \pm 0.57$ | $50.21 \pm 3.33$ |
| Magic | STaSy | $77.52 \pm 0.27$ | - | $92.92 \pm 0.30$ | $91.18 \pm 1.30$ | $46.07 \pm 1.61$ | $88.12 \pm 7.05$ | $90.27 \pm 6.53$ | $75.02 \pm 4.03$ | $52.57 \pm 0.92$ |
| | STaSy+**IJF** | $72.52 \pm 6.59$ | **6.45%** ↓ | $90.90 \pm 0.59$ | $90.54 \pm 1.86$ | $24.11 \pm 0.86$ | $89.21 \pm 1.28$ | $84.58 \pm 0.49$ | $75.70 \pm 5.89$ | $50.79 \pm 1.14$ |
| | STaSy+**TCM** | $75.12 \pm 0.29$ | **3.10%** ↓ | $91.49 \pm 0.63$ | $92.50 \pm 3.01$ | $35.24 \pm 1.48$ | $89.62 \pm 5.33$ | $89.96 \pm 6.44$ | $75.70 \pm 5.53$ | $49.85 \pm 0.21$ |
| | STaSy+**TCMP** | $76.70 \pm 0.38$ | **1.06%** ↓ | $92.77 \pm 0.20$ | $97.27 \pm 1.30$ | $40.11 \pm 1.65$ | $95.37 \pm 1.51$ | $96.34 \pm 0.42$ | $76.63 \pm 6.85$ | $48.41 \pm 0.28$ |
| | TabDDPM | $77.62 \pm 2.11$ | - | $92.78 \pm 0.23$ | $98.41 \pm 0.37$ | $46.67 \pm 1.18$ | $99.07 \pm 0.06$ | $98.58 \pm 0.51$ | $99.05 \pm 0.70$ | $50.47 \pm 0.42$ |
| | TabDDPM+**IJF** | $63.54 \pm 1.55$ | **18.15%** ↓ | $90.20 \pm 0.52$ | $91.81 \pm 0.67$ | $21.05 \pm 1.48$ | $93.69 \pm 0.73$ | $84.60 \pm 0.78$ | $95.90 \pm 3.00$ | $50.47 \pm 0.86$ |
| | TabDDPM+**TCM** | $72.99 \pm 0.22$ | **5.96%** ↓ | $91.69 \pm 0.86$ | $97.92 \pm 0.38$ | $32.51 \pm 0.70$ | $98.97 \pm 0.08$ | $99.19 \pm 0.11$ | $97.62 \pm 2.44$ | $49.73 \pm 0.35$ |
| | TabDDPM+**TCMP** | $76.22 \pm 0.39$ | **1.81%** ↓ | $91.50 \pm 0.22$ | $96.50 \pm 4.02$ | $36.52 \pm 2.43$ | $98.08 \pm 1.32$ | $95.17 \pm 3.05$ | $95.29 \pm 6.22$ | $49.88 \pm 0.74$ |
| | TabSyn | $80.02 \pm 0.39$ | - | $93.18 \pm 0.31$ | $99.10 \pm 0.68$ | $48.28 \pm 0.41$ | $99.00 \pm 0.28$ | $99.15 \pm 0.08$ | $99.75 \pm 0.29$ | $50.48 \pm 0.16$ |
| | TabSyn+**IJF** | $57.03 \pm 5.52$ | **28.73%** ↓ | $90.41 \pm 1.65$ | $90.52 \pm 1.70$ | $21.43 \pm 3.77$ | $92.89 \pm 0.16$ | $84.95 \pm 0.67$ | $95.73 \pm 3.80$ | $50.05 \pm 0.76$ |
| | TabSyn+**TCM** | $52.06 \pm 7.12$ | **34.94%** ↓ | $91.77 \pm 0.12$ | $96.83 \pm 0.40$ | $30.79 \pm 2.92$ | $97.83 \pm 0.65$ | $98.09 \pm 0.17$ | $93.55 \pm 1.49$ | $51.76 \pm 0.49$ |
| | TabSyn+**TCMP** | $76.46 \pm 0.36$ | **4.44%** ↓ | $91.91 \pm 0.42$ | $98.03 \pm 1.76$ | $39.54 \pm 1.54$ | $98.87 \pm 0.57$ | $97.26 \pm 0.27$ | $97.58 \pm 3.36$ | $51.32 \pm 0.63$ |

*Table 9.* The overview performance comparison for tabular diffusion models on more generative models. "TCM" represents our proposed **TabCutMix** and "TCMP" represents **TabCutMixPlus**. "Mem. Ratio" represents memorization ratio. "Improv" represents the improvement ratio on memorization.

| | Methods | Mem. Ratio (%)↓ | **Improv.** | MLE (%)↑ | α-Precision(%)↑ | β-Recall(%)↑ | Shape Score(%)↑ | Trend Score(%)↑ | C2ST(%)↑ | DCR(%) |
|---|---|---|---|---|---|---|---|---|---|---|
| Default | CTGAN | 12.83 ± 0.63 | - | 68.68 ± 0.21 | 68.95 ± 1.70 | 16.49 ± 0.55 | 85.04 ± 0.93 | 77.05 ± 2.47 | 61.89 ± 4.89 | 49.45 ± 0.80 |
| | CTGAN+**TCMP** | 11.80 ± 0.25 | **8.03% ↓** | 70.05 ± 0.81 | 71.33 ± 0.82 | 17.02 ± 0.60 | 85.28 ± 0.97 | 78.09 ± 0.32 | 63.67 ± 3.07 | 50.14 ± 0.60 |
| | TVAE | 17.22 ± 0.43 | - | 72.24 ± 0.36 | 82.97 ± 0.37 | 20.57 ± 0.42 | 89.37± 0.54 | 83.36 ± 0.99 | 52.63 ± 0.37 | 51.33 ± 0.96 |
| | TVAE+**TCMP** | 15.59 ± 0.28 | **9.47% ↓** | 72.75 ± 0.52 | 81.57 ± 0.32 | 19.52 ± 0.39 | 89.33 ± 0.48 | 78.04 ± 6.05 | 45.60 ± 0.90 | 50.07 ± 0.52 |
| Adult | CTGAN | 21.68 ± 0.62 | - | 88.64 ± 0.32 | 78.15 ± 3.66 | 26.27 ± 0.67 | 82.43 ± 0.90 | 82.83 ± 0.93 | 63.64 ± 2.74 | 49.14 ± 0.27 |
| | CTGAN+**TCMP** | 21.20 ± 0.31 | **2.23% ↓** | 88.63 ± 0.66 | 76.27 ± 0.60 | 25.56 ± 0.33 | 81.42 ± 0.32 | 82.11 ± 1.01 | 60.98 ± 1.67 | 50.96± 0.17 |
| | TVAE | 30.78 ± 0.41 | - | 88.61 ± 0.49 | 92.34 ± 2.19 | 29.90 ± 1.00 | 82.71 ± 0.45 | 79.06 ± 0.90 | 48.67 ± 2.70 | 48.76 ± 0.25 |
| | TVAE+**TCMP** | 29.88 ± 0.34 | **2.93% ↓** | 88.42 ± 0.26 | 88.02 ± 0.72 | 29.78 ± 1.42 | 82.76 ± 0.92 | 77.92 ± 0.66 | 52.52± 7.24 | 51.35 ± 0.29 |
| Shoppers | CTGAN | 19.80 ± 1.15 | - | 83.22 ± 1.30 | 85.26 ± 5.17 | 26.67 ± 1.66 | 77.73 ± 0.50 | 86.48 ± 0.69 | 67.18± 5.84 | 48.54 ± 0.74 |
| | CTGAN+**TCMP** | 19.69 ± 0.35 | **0.52% ↓** | 83.94 ± 0.80 | 82.07 ± 9.62 | 23.47 ± 0.43 | 76.15 ± 0.22 | 84.56 ± 0.40 | 64.60 ± 0.82 | 52.02 ± 0.13 |
| | TVAE | 23.17 ± 0.77 | - | 86.65 ± 0.48 | 50.32 ± 2.98 | 11.16 ± 1.33 | 74.75 ± 0.92 | 77.56 ± 1.11 | 21.29 ± 3.28 | 43.31 ± 0.65 |
| | TVAE+**TCMP** | 20.16 ± 0.80 | **12.98% ↓** | 87.07 ± 0.74 | 50.69 ± 0.44 | 10.54 ± 1.68 | 74.86 ± 0.26 | 78.55± 0.48 | 21.78 ± 4.28 | 43.21 ± 0.69 |
| Magic | CTGAN | 74.08 ± 0.69 | - | 83.62 ± 0.33 | 82.96 ± 0.83 | 8.70 ± 0.69 | 89.65 ± 0.45 | 91.81± 1.18 | 63.18 ± 0.53 | 51.53 ± 2.07 |
| | CTGAN+**TCMP** | 69.12 ± 1.94 | **6.70% ↓** | 82.21 ± 0.43 | 82.72± 1.19 | 8.52 ± 0.76 | 90.70 ± 1.40 | 89.07 ± 0.16 | 65.99 ± 10.68 | 49.37 ± 0.31 |
| | TVAE | 77.20 ± 0.32 | - | 88.71 ± 0.45 | 92.17 ± 0.43 | 32.11 ± 0.81 | 91.67 ± 0.75 | 93.75 ± 0.43 | 77.18 ± 1.38 | 48.71 ± 0.59 |
| | TVAE+**TCMP** | 70.26 ± 0.97 | **8.98% ↓** | 87.36 ± 0.51 | 91.31 ± 1.33 | 26.17 ± 0.83 | 89.01 ± 0.79 | 92.06 ± 0.23 | 73.95 ± 1.79 | 53.14 ± 0.52 |

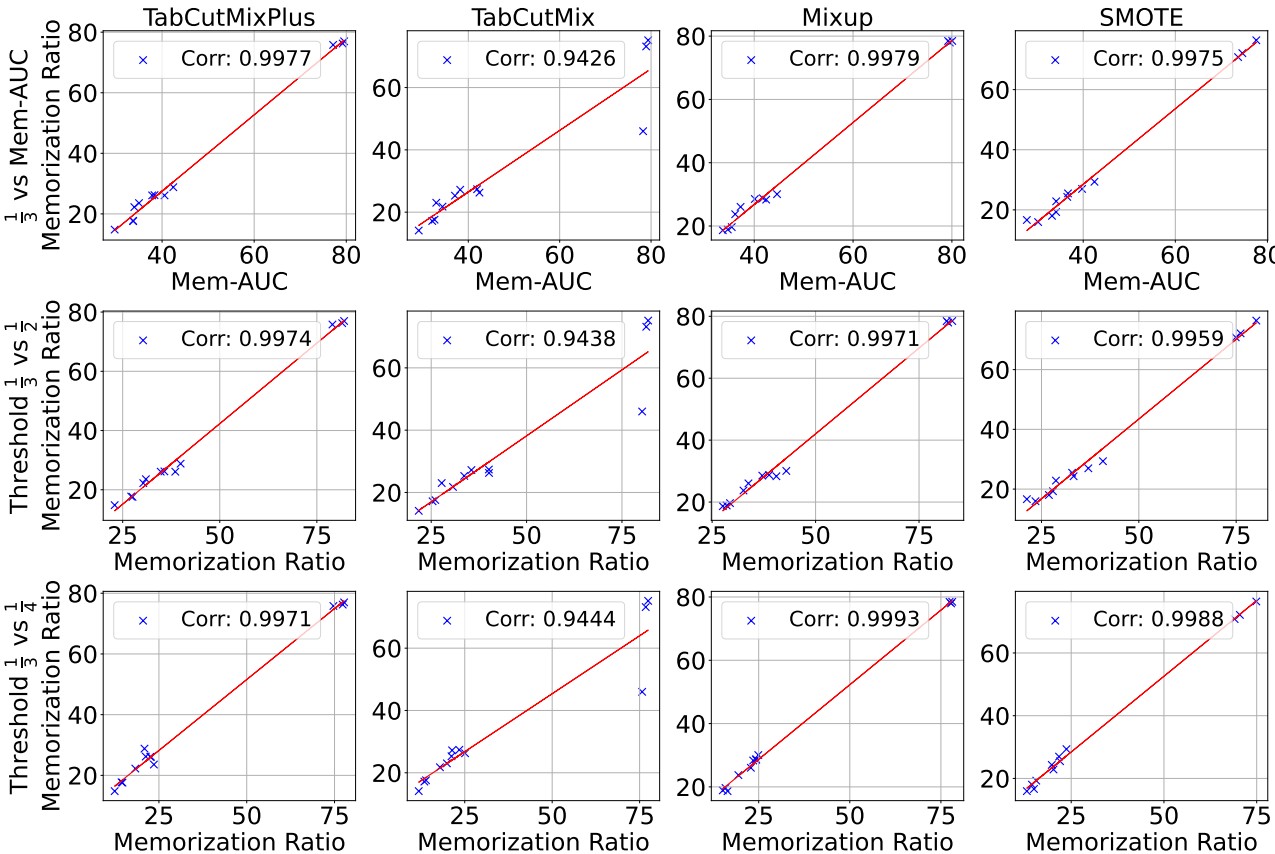

*Figure 11.* Correlation analysis between the memorization ratio with threshold 1/3 and Mem-AUC and the memorization ratio with different thresholds.

*Table 10.* Mem-Ratio vs. Mem-AUC

| Methods | Metrics | Default | | | Magic | | | Shoppers | | | Adult | | |
|---|---|---|---|---|---|---|---|---|---|---|---|---|---|
| | | TabSyn | TabDDPM | STaSy | TabSyn | TabDDPM | STaSy | TabSyn | TabDDPM | STaSy | TabSyn | TabDDPM | STaSy |
| TabCutMixPlus | Mem-Ratio | 17.60 | 17.73 | 14.81 | 76.26 | 75.80 | 76.99 | 26.11 | 28.81 | 23.57 | 26.20 | 26.10 | 22.24 |
| | Mem-AUC | 33.73 | 33.70 | 29.71 | 79.21 | 77.15 | 79.53 | 37.85 | 42.44 | 34.99 | 38.36 | 40.54 | 33.98 |
| TabCutMix | Mem-Ratio | 17.59 | 17.17 | 14.11 | 45.99 | 73.19 | 75.18 | 25.30 | 26.27 | 23.02 | 27.20 | 27.37 | 21.78 |
| | Mem-AUC | 32.61 | 32.10 | 29.15 | 78.26 | 78.93 | 79.37 | 37.08 | 42.44 | 32.99 | 38.23 | 41.93 | 34.35 |
| Mixup | Mem-Ratio | 19.66 | 18.89 | 18.66 | 78.45 | 78.48 | 77.97 | 28.54 | 28.76 | 26.04 | 28.33 | 30.06 | 23.73 |
| | Mem-AUC | 35.45 | 34.65 | 33.64 | 79.24 | 80.07 | 79.57 | 40.12 | 41.74 | 37.24 | 42.42 | 44.62 | 36.15 |
| SMOTE | Mem-Ratio | 19.29 | 18.02 | 15.95 | 70.81 | 72.14 | 76.42 | 25.48 | 26.63 | 22.85 | 26.96 | 29.34 | 24.31 |
| | Mem-AUC | 34.12 | 33.23 | 30.21 | 73.63 | 74.55 | 77.58 | 36.70 | 37.77 | 34.14 | 39.76 | 42.45 | 36.54 |

during augmentation.

- Challenges in Complex Domains. TabCutMix struggles in sensitive domains, such as healthcare or finance, where feature interactions often carry critical domain-specific meanings. Arbitrary feature exchanges may result in implausible or nonsensical combinations, reducing the utility of augmented data for downstream tasks. For example, relationships between features like age and medical diagnosis may be violated, leading to unrealistic augmented samples.

- Classification-specific applicability: TabCutMix and TabCutMixPlus are specifically designed for classification tasks, where samples can be grouped based on discrete class labels. This design makes it challenging to directly apply the methods to regression tasks, where the target variable is continuous and lacks discrete boundaries for grouping. Future work could explore strategies such as pseudo-labeling, binning continuous targets, or developing regression-aware augmentation techniques to adapt the core ideas of TabCutMix to regression settings and expand the scope of its applicability.

- Sensitivity to Outliers. The proposed mixed-distance metric, like many distance-based measures, is sensitive to outliers, particularly in numerical features, which can disproportionately affect distance calculations and distort relationships between samples. While normalization mitigates feature dominance, it does not fully address the impact of extreme values. Future work could explore robust distance metrics, such as adaptive scaling or trimming, to reduce the influence of outliers and improve the reliability of distance-based approaches in tabular data modeling.

- Lack of General Insights into Data-Centric Factors. While this work identifies the influence of data-centric factors, such as dataset complexity and feature interactions, on the effectiveness of TabCutMix, it does not provide a comprehensive framework for understanding or addressing these factors. A deeper investigation into these data-centric elements is necessary to fully realize the potential of TabCutMix and similar augmentation techniques.

# G. Future Work

While this study makes significant strides in addressing the issue of memorization in tabular data generation, several directions remain open for future exploration:

1. **Theoretical Analysis of Factor Heterogeneity**: A deeper theoretical investigation is valuable into how different factors, such as dataset size and feature dimensionality, influence memorization in heterogeneous ways. Specifically, understanding the nonlinear relationships between these factors and their combined effect on model memorization could provide further insights.

2. **Exploring Alternative Memorization Mitigation Techniques**: Beyond data augmentation, future work could explore different strategies to mitigate memorization. These could include **more advanced generative model training techniques**, such as regularization methods or differential privacy mechanisms that limit model overfitting to specific data points. Additionally, techniques like **model pruning, weight clipping, or dropout variations** could be explored for their potential to reduce memorization during model training. The model architecture design by leveraging architectures like variational autoencoders (VAEs), normalizing flows, or GAN variations with modified loss functions can be developed to mitigate memorization. The prior information of the flag to the model to indicate whether it is processing real or augmented data can also integrated for advanced memorization mitigation method.

3. **Evaluation Metrics for Memorization**: Developing more comprehensive and practical evaluation metrics specifically tailored to detect memorization in tabular data models remains a key area for future work. These metrics could better assess the trade-off between generating high-quality synthetic data and avoiding overfitting to the training data.

4. **Real-World Applications and Use Cases**: Applying these methods to a broader range of real-world use cases could provide valuable feedback and improvements. Specific industries such as healthcare, finance, and marketing, where tabular data is prevalent, would be ideal candidates for testing how well these approaches generalize and perform in production environments.

5. **Data-Centric Investigation for Tabular Diffusion Models**: This study reveals that memorization in tabular diffusion models may be predominantly driven by dataset-specific factors such as feature complexity, sparsity, and redundancy, rather than model-specific architecture. A deeper exploration into these data-centric influences could provide valuable insights into the interplay between dataset properties and memorization behavior. Future work could focus on developing strategies to quantify the impact of dataset characteristics on generative performance and proposing adaptive preprocessing, augmentation, or sampling methods tailored to diverse datasets. Such investigations would enhance the robustness and applicability of diffusion models across a wide range of tabular data scenarios.

6. **Cross-Domain Generalization and Transferability of Memorization Mitigation**: The generalization of memorization insights and mitigation strategies developed for tabular diffusion models to other data modalities (e.g., text, images, multimodal data) and diverse generative architectures (e.g., VAEs, GANs, autoregressive models) are under-explored. This includes investigating shared factors (e.g., data sparsity, complexity) and domain-specific drivers of memorization to establish unified detection frameworks and evaluating the adaptability of techniques like TabCutMix and TabCutMixPlus across models and domains. Empirical studies will assess their effectiveness in reducing memorization while preserving generation quality, and adaptive variants tailored to specific architectures (e.g., sequential text models) or data characteristics (e.g., high-dimensional images) will be developed to broaden real-world applicability. Theoretical comparisons of memorization behaviors across heterogeneous data types will further clarify universal principles and context-dependent challenges.

