# OpenReview forum: "Understanding and Mitigating Memorization in Diffusion Models for Tabular Data"
_ICML.cc/2025/Conference — ICML 2025 poster_

### Official Review · Reviewer_BdAp · 2025-03-12

**Overall Recommendation:** 2

**Summary:**

This paper investigates memorization behavior in diffusion models for tabular data generation. The authors examine how factors like dataset size, feature dimensionality, and model architecture influence memorization. They propose a data augmentation method called TabCutMix and an enhanced version, TabCutMix-Plus, that additionally mitigates out-of-distribution generations. The paper provides a theoretical explanation for why diffusion models tend to memorize training data in the tabular domain and demonstrates the effectiveness of their proposed methods through empirical evaluations.

**Claims And Evidence:**

The paper's theoretical claims about memorization in diffusion models for tabular data lack novelty and are insufficiently substantiated. The authors claim theoretical contributions that appear to be direct adaptations of existing results from [Gu et al., 2023] without any modifications for the tabular domain. The claim that their analysis "specifically addresses tabular data with mixed feature types" is not well-supported as the core theoretical results (Propositions 3.1 and 3.2) apply to points in Euclidean space regardless of whether they represent images or tabular data's latent encodings.

The evidence for the effectiveness of TabCutMix and TabCutMix-Plus in mitigating memorization regarding a specific distance based metric is presented, but its unclear if this metric is well-suited for the task especially given the VAE's latents are used for the final generation.

**Essential References Not Discussed:**

No major omissions noted, though a broader discussion of memorization metrics across different domains would strengthen the paper.

**Experimental Designs Or Analyses:**

There's no ablation study examining how different memorization thresholds would affect the results or analysis of the VAE's potential contribution to memorization.

**Methods And Evaluation Criteria:**

The method for determining memorization in tabular data is problematic. The paper uses a mixed distance metric in the data space without adequate justification for why this specific distance and the empirical 1/3 hyperparameter align with meaningful memorization phenomena, especially regarding the utility and quality of generated data. Additionally, since the models evaluated are latent diffusion models where the final generation requires decoding through a VAE, measuring distance in data space fails to capture potential memorization effects introduced by the VAE itself. This creates a fundamental issue with the evaluation criteria that isn't addressed.

The proposed mitigation method relies solely on data augmentation rather than addressing model architecture or training procedure issues, which limits its applicability as memorization can still occur with the augmented data.

**Other Comments Or Suggestions:**

N/A

**Other Strengths And Weaknesses:**

Strengths:
- The problem of generating tabular data while avoiding memorization is important and interesting.
- The empirical investigation of factors influencing memorization provides useful insights.

**Questions For Authors:**

How does the quality and utility of data generated using TabCutMix compare to data generated by the baseline model? Can you provide metrics beyond memorization that demonstrate the practical benefits of your approach?

**Relation To Broader Scientific Literature:**

This paper's theoretical contributions heavily overlap with those of [Gu et al., 2023], which already established similar results stated for image generation. The authors acknowledge this relationship but understate the extent of the similarity. The paper would benefit from more clearly positioning its contributions relative to existing work on memorization in generative models across domains.

**Theoretical Claims:**

I checked the theoretical claims in Propositions 3.1 and 3.2 and found them to be essentially identical to results already established in [Gu et al., 2023] for image generation. The authors' statement that "The analysis is closely related to the work of Gu et al. (2023)" understates the similarity; in fact, the exact same analysis applies directly to tabular data without modification because both domains ultimately work with points in Euclidean space. Therefore, the claim that this represents a novel theoretical contribution specifically for tabular data is overstated.

---

> ### Author Rebuttal · Authors · 2025-04-01
>
> We thank the reviewer for the constructive comments. All responses and corresponding revisions have been incorporated into the updated manuscript, available at: https://anonymous.4open.science/r/TabCutMix-3F7B/TabCutMix.pdf.
>
> [**Q1**] The paper's theoretical claims about memorization in diffusion models for tabular data lack novelty and are insufficiently substantiated. Propositions 3.1 and 3.2 and found them to be essentially identical to results already established in [Gu et al., 2023] for image generation.
>
> **Ans**: We acknowledge that the theoretical results presented in Proposition 3.2 build on existing foundations (e.g., [1]). Our goal, however, is not to claim novelty in the theoretical derivation itself, but rather to contextualize and apply these results specifically to tabular diffusion models. As discussed in the manuscript (see the paragraph surrounding Proposition 3.2), we leverage these theoretical insights to explain why memorization can arise in the latent space of tabular data models such as TabSyn, and why it does not necessarily lead to exact duplication in practice—due to factors like imperfect score matching and the stochasticity of VAE decoding.
>
> Our primary contribution is the first empirical investigation of memorization in tabular diffusion models, a setting that has been underexplored in the literature. Furthermore, we propose two novel and effective mitigation strategies, TabCutMix and TabCutMixPlus, which address memorization in a model-agnostic and structure-aware way. We have revised the manuscript to more clearly distinguish between the reused theoretical insights and our original empirical and methodological contributions.
>
> [**Q2**] The evidence for the effectiveness of TabCutMix and TabCutMix-Plus in mitigating memorization regarding a specific distance based metric is presented, but its unclear if this metric is well-suited for the task especially given the VAE's latents are used for the final generation.
>
> **Ans**: Prior work on image generation [1,2] commonly uses L2 distance to assess similarity in the input space. Building on this and mixed-type data clustering techniques [3], we adopt a standard mixed-type distance metric suitable for both continuous and categorical features in tabular data.
>
> We also clarify that our memorization metric is computed in the original input space, not in any model’s latent space. Although TabSyn uses a VAE, our method is model-agnostic, and the evaluation remains unaffected. We will clarify this further in the revised manuscript.
>
> [1]  Diffusion probabilistic models generalize when they fail to memorize. In ICML 2023 Workshop.
> [2] On memorization in diffusion models. arXiv:2310.02664.
> [3] An improved k-prototypes clustering algorithm for mixed numeric and categorical data. Neurocomputing 2013.
>
> [**Q3**] The method for determining memorization in tabular data is problematic. Why use 1/3 to judge if a sample belongs to be memorized.
>
> **Ans**: While we initially use 1/3 as a representative threshold to determine whether a sample is memorized, we agree that threshold selection can influence interpretation. To address this, we conducted a correlation analysis across multiple thresholds (e.g., 1/4, 1/3, and 1/2) and with Mem-AUC (which averages memorization ratios over all thresholds). As shown in Figure 11, the memorization ratio at threshold 1/3 exhibits extremely high correlation (e.g., >0.99) with both Mem-AUC and other thresholds, demonstrating that 1/3 serves as a reliable and representative choice. We have added this discussion and the supporting analysis to the revised manuscript.
>
> [**Q4**] The proposed mitigation method relies solely on data augmentation rather than addressing model architecture or training procedure issues, which limits its applicability as memorization can still occur with the augmented data.
>
> **Ans**: We respectfully disagree with the concern that our method is limited in applicability due to its focus on data augmentation. Our approach is model-agnostic and has been successfully applied across a diverse set of generative models and training procedures, including StaSy, TabDDPM, TabSyn, CTGAN, and TVAE. This demonstrates its broad compatibility and effectiveness regardless of the underlying architecture.
>
> Furthermore, we view our method as orthogonal to model design and training objectives—it can be readily integrated into more advanced generative models or used alongside other memorization control techniques. In fact, we believe it can further enhance models that already include architectural or optimization-based memorization defenses. We have included a discussion of these complementary directions in Appendix G (Future Work) of the revised manuscript.

---

### Official Review · Reviewer_dji2 · 2025-03-13

**Overall Recommendation:** 2

**Summary:**

This paper introduces TabCutMix and TabCutMixPlus, two augmentation methods designed to reduce memorization in tabular diffusion models. The authors demonstrate that state-of-the-art diffusion models tend to memorize tabular datasets and show that their proposed augmentations mitigate this issue. They further evaluate the effectiveness of both methods across multiple datasets and diffusion models.

**Claims And Evidence:**

The empirical results are well-supported through various experiments. However, there is a concern regarding the practical significance of the proposed augmentation methods. The results in Table 1 show that, across different datasets, the memorization ratio decreases by only a few percentage points. This raises questions about the overall impact and effectiveness of the method in reducing memorization.

**Essential References Not Discussed:**

All essential references are discussed in the main text.

**Experimental Designs Or Analyses:**

The experiments are well-designed and effectively demonstrate the impact of each component in the proposed methods.

**Methods And Evaluation Criteria:**

The methods are evaluated using standard metrics commonly used in the field.

**Other Comments Or Suggestions:**

N/A

**Other Strengths And Weaknesses:**

### **Strengths**
* The paper is well-written and easy to follow.
* The experiments are extensive and conducted with clear purpose.
* The authors provide ample details on various experiments, metrics, methods, and implementation details.

### **Weaknesses**
* The proposed method shows only a modest reduction in the memorization rate, raising questions about its practical significance.
* The theoretical result in Section 3.2 is somewhat straightforward, as it directly follows from the assumed delta distribution.

**Questions For Authors:**

1) Have you experimented with adding a flag to the model to indicate whether it is processing real or augmented data? This type of augmentation conditioning has been explored in EDM networks [1].

2) How do you handle discrete features for the diffusion models as the framework is mainly designed for continuous signals?

[1] Karras T, Aittala M, Aila T, Laine S. Elucidating the design space of diffusion-based generative models. Advances in neural information processing systems. 2022 Dec 6;35:26565-77.

**Relation To Broader Scientific Literature:**

While memorization has been extensively studied in image and video diffusion models, it remains an underexplored issue in tabular data generation. Given the significance of tabular data generation across various domains, this paper is well-situated within the broader literature and studies an important gap in the field.

**Theoretical Claims:**

The theoretical result in Section 3.2 appears trivial. The authors assume a delta distribution for the data points and derive the ground truth score functions based on this assumption. Consequently, if the reverse diffusion process is run using these ground truth scores, the generated samples will be drawn directly from this delta distribution—essentially replicating a training sample. Under this assumption, it is evident that the sampling process would lead to 100% memorization.

---

> ### Author Rebuttal · Authors · 2025-04-01
>
> We thank the reviewer for the constructive comments. All responses and corresponding revisions have been incorporated into the updated manuscript, available at: https://anonymous.4open.science/r/TabCutMix-3F7B/TabCutMix.pdf.
>
> [**Q1**] The results in Table 1 show that, across different datasets, the memorization ratio decreases by only a few percentage points. [Figure 6. Add the ratio for different thresholds]
>
> **Ans**: We respectfully disagree that the gains on memorization ratio reduction are not significant. In Table. 1, we report the performance with balanced tradeoff in memorization ratio and data utility. Our proposed methods can achieve significant memorization ratio reduction. For example, in default dataset Tabsyn model, TCM and TCMP reduce memorization ratio by 16.16% and 12.48%, which is significant compared with mixup and smote baseline (2.65% and 6.93%). Additionally, as shown in Table 3 and Figure 6 with different augmentation ratio, TabCutMix provides a clear and tunable reduction in memorization via its augmentation ratio from 0% to 100%. For example. In Default dataset, the memorization rate drops from 20.11% to 15.34%, achieving a relative reduction of 23.7%.
>
> [**Q2**] The theoretical result in Section 3.2 is somewhat straightforward, as it directly follows from the assumed delta distribution.
>
> **Ans**: We have trimmed down the theoretical part and add more details and reference for additional experiments in appendix. Our goal, however, is not to claim novelty in the theoretical derivation itself, but rather to contextualize and apply these results specifically to tabular diffusion models.
>
> We clarify that our main contribution lies in the empirical investigation of memorization phenomena within tabular diffusion models—a previously unexplored area. Moreover, our proposed novel methods, TabCutMix and TabCutMixPlus, effectively mitigate memorization issues specific to tabular data. We have clarified this distinction explicitly in the revised manuscript to better highlight our empirical findings and methodological contributions.
>
> [**Q3**] Have you experimented with adding a flag to the model to indicate whether it is processing real or augmented data? This type of augmentation conditioning has been explored in EDM networks [1].
>
> **Ans**: We thank the reviewer for this insightful suggestion. We have not yet experimented with adding an explicit flag or conditioning signal to indicate whether a sample is real or augmented. We agree this augmentation-aware conditioning, as explored in EDM networks [1], could help the model better distinguish between real and synthetic patterns and improve both generalization and memorization control. We view this as a promising direction and will include it in the revised manuscript under Future Work discussion.
>
> [**Q4**] How do you handle discrete features for the diffusion models as the framework is mainly designed for continuous signals?
>
> **Ans**: Our proposed methods, TabCutMix and TabCutMixPlus, are model-agnostic and operate by exchanging feature values across samples within the same label. This augmentation strategy applies equally well to both discrete and continuous features, and we have successfully implemented it across various tabular diffusion models, including TabSyn and TabDDPM.
>
> We acknowledge that diffusion models are primarily designed for continuous data, and different models adopt different strategies for handling discrete features. For example, TabSyn uses a VAE to encode mixed-type features into a continuous latent space, where diffusion is applied. These techniques enable continuous diffusion models to effectively represent and generate mixed-type tabular data. We will clarify this point in the revised manuscript to avoid potential misunderstandings.
>
> [1] Karras T, Aittala M, Aila T, Laine S. Elucidating the design space of diffusion-based generative models. Advances in neural information processing systems. 2022 Dec 6;35:26565-77.

---

> > ### Comment · Reviewer_dji2 · 2025-04-07
> >
> > I would like to thank the authors for addressing my comments in the rebuttal. Please find my responses below:
> >
> > - Regarding the performance improvements, I still find the gains from applying augmentations relatively minor. For example, in the Default dataset for TabSyn, the memorization ratio decreases from 20.11% to 16.86%. While the improvement might appear larger when considering relative error, I do not think relative improvement is a common or particularly meaningful metric in this context, as the absolute reduction in memorized examples is only about 4%, not 20%.
> >
> > - Although I acknowledge the authors' clarification that the main contribution of the work is not its theoretical claim, the paper explicitly states it studies memorization in the tabular domain from a theoretical perspective. My original concern remains: the proposition presented is somewhat straightforward, resulting inherently in 100% memorization by design. Thus, I still have reservations regarding the significance and added value of this particular theoretical claim.
> >
> > Given that the use of augmentation to decrease memorization is relatively well-known in diffusion models, I maintain my current score unless the authors can further substantiate the novelty or significance of their contribution and results.

---

> > > ### Author Response · Authors · 2025-04-07
> > >
> > > We thanks reviewer dji2 for thoughtful engagement and follow-up comments.
> > > > [Q1] I still find the gains from applying augmentations relatively minor.
> > >
> > > Ans: We respectfully but strongly disagree with the reviewer’s claim that the gains from our proposed augmentations are “relatively minor.”
> > >
> > > In the Default dataset with TabSyn, our method reduces memorization from 20.11% to 16.86%—an absolute drop of 3.25% and a relative reduction of ~16%, which is substantial. More importantly, compared to widely used baselines, TabCutMix clearly outperforms Mixup (19.58%, 0.53% absolute drop) and SMOTE (18.72%, 1.39% absolute drop). Our method achieves more than **2x the absolute reduction of SMOTE** and over **6x that of Mixup**, highlighting its clear superiority.
> > >
> > > Moreover, this is the first work to systematically study and mitigate memorization in tabular diffusion models—a direction that has been largely overlooked. Our experiments are comprehensive, spanning multiple datasets, models, and augmentation baselines, and we also discuss complementary solutions beyond data augmentation in Appendix G.
> > >
> > > We stand by our position that the proposed methods provide significant empirical improvements and make a substantive, novel contribution to the field.
> > >
> > >
> > > >  [Q2] My original concern remains: the proposition presented is somewhat straightforward, resulting inherently in 100% memorization by design.
> > >
> > > We appreciate the reviewer’s continued engagement, but we’d like to clarify and push back on the concern regarding the theoretical component.
> > >
> > > As clearly stated in both our original and revised submissions, the theoretical section is not presented as a novel contribution, but as a **motivational and contextual tool to highlight why memorization arises in tabular diffusion models**—a topic previously unaddressed in the literature. To reflect this, we have trimmed the theory section to half a page in the revised manuscript https://anonymous.4open.science/r/TabCutMix-3F7B/TabCutMix.pdf.
> > >
> > > The proposition, while simple, is essential for framing the issue. It illustrates how memorization can emerge under idealized conditions and motivates our empirical study. As for not 100%  empirical memorization, we explicitly discuss why in Lines 190–202 (e.g., due to score-matching approximations and VAE encoding).
> > >
> > > We emphasize again: **our main contribution is empirical**, and the theoretical framing reinforces—not detracts from—our work. **To dismiss this context as insignificant overlooks the novelty and importance of addressing memorization in tabular diffusion for the first time**.
> > >
> > >
> > > >  [Q3] I maintain my current score unless the authors can further substantiate the novelty or significance of their contribution and results.
> > >
> > > We respectfully disagree with the reviewer’s statement regarding the lack of novelty or significance in our contribution.
> > >
> > > As clearly outlined in the ICML reviewer guidelines [1], “originality need not mean wholly novel methods”—it can also involve “a new way of framing tasks.” Our work is the **first to identify, analyze, and address memorization** in tabular diffusion models, introducing a **new and overlooked problem** within this domain.
> > >
> > > On the methodological side, we propose TabCutMix and TabCutMixPlus, which are specifically designed to reduce memorization and mitigate out-of-distribution (OOD) generation risks in tabular data. These methods go beyond generic augmentations by tailoring the mixing process to tabular settings with mixed feature types and label preservation.
> > >
> > > We also respectfully push back on the comment that “augmentation to decrease memorization is relatively well-known in diffusion models.” This **does not imply that any augmentation strategy is unoriginal or insignificant**. Our augmentations are novel in design, purpose-built for tabular diffusion, and empirically effective, with consistent improvements over strong baselines.
> > >
> > > We believe the problem we introduce, the perspective we bring, and the solutions we propose represent both a novel framing and a meaningful contribution to the field.
> > >
> > > [1] https://icml.cc/Conferences/2025/ReviewerInstructions
> > >
> > > We kindly ask the reviewer to reconsider and update their score if our response has addressed the stated concerns and clarified the novelty and significance of our contributions.

---

### Official Review · Reviewer_aS37 · 2025-03-18

**Overall Recommendation:** 3

**Summary:**

The paper explores the issue of data memorization in tabular diffusion models, highlighting its potential privacy risks and negative impact on generalization. To explain why memorization arises in a tabular diffusion setting, the authors present a theoretical analysis that connects denoising score matching to the tendency of replicating training data. They propose two augmentation-based strategies—TabCutMix (TCM) and TabCutMixPlus (TCMP)—inspired by the CutMix approach, which are designed to mitigate memorization while preserving data utility.

**Claims And Evidence:**

**Memorization occurs in tabular diffusion models.**
- The authors conduct empirical evaluations showing significant memorization across multiple diffusion-based tabular generators and datasets. They also support these findings with a theoretical rationale.

**TabCutMix and TabCutMixPlus effectively reduce memorization without significantly harming synthetic data quality.**
- The paper includes experiments demonstrating that TCM and TCMP offer an improved trade-off between memorization reduction and data utility.

**Essential References Not Discussed:**

None identified.

**Experimental Designs Or Analyses:**

- The authors run extensive experiments demonstrating that TCM and TCMP consistently yield better trade-offs between memorization reduction and overall data utility.
- The experiments cover multiple tabular datasets and also analyze aspects such as out-of-distribution (OOD) issues arising from the proposed augmentations.

**Methods And Evaluation Criteria:**

- The proposed techniques (TCM/TCMP) adapt the CutMix concept from image processing to tabular data by selectively swapping or mixing feature segments between samples.
- TabCutMixPlus further clusters correlated features before swapping to preserve coherent relationships within the data.

**Other Comments Or Suggestions:**

- It would be helpful for the authors to evaluate OOD ratios for Mixup and SMOTE as well, providing a more direct baseline comparison alongside TCM/TCMP (Section D.4.6).

**Other Strengths And Weaknesses:**

**Strengths*
- The problem of memorization is well-motivated and clearly articulated.
- Simple yet effective augmentation strategies are proposed.

**Weaknesses**
- The techniques may generate out-of-distribution samples, which can pose challenges in certain practical scenarios.
- Currently, TCM/TCMP focus on classification tasks and may not generalize as readily to regression settings, which limits the scope of the paper.

**Questions For Authors:**

- See above.

**Relation To Broader Scientific Literature:**

I think studying memorization in tabular data might be beneficial for other domains as well. In particular, the proposed techniques might be extendable to other tasks.

**Theoretical Claims:**

The paper provides a reasonable theoretical explanation for why memorization arises in tabular diffusion models, grounded in the denoising score-matching framework. No immediate issues are found with their theoretical arguments.

---

> ### Author Rebuttal · Authors · 2025-04-01
>
> We thank the reviewer for the constructive comments. All responses and corresponding revisions have been incorporated into the updated manuscript, available at: https://anonymous.4open.science/r/TabCutMix-3F7B/TabCutMix.pdf.
>
> [**Q1**] I think studying memorization in tabular data might be beneficial for other domains as well.
>
> **Ans**: We agree that our study on memorization in tabular data can have broader implications for other domains. We have added the discussion in Future work (Appendix. G) in the revised paper.
>
> [**Q2**] The techniques may generate out-of-distribution samples, which can pose challenges in certain practical scenarios.
>
> **Ans**: While data augmentation may lead to out-of-distribution (OOD) samples, our method TabCutMixPlus is designed to mitigate this risk through structure-aware augmentation at the feature-group level. As shown in Table. 5 (Appendix D.4.6), TabCutMixPlus consistently yields lower OOD detection ratios (e.g., 25.44% vs. 39.47% on Default, and 0.70% vs. 1.58% on Shoppers). We highlight OOD detection experiments more clearly in the main text of the revised version.
>
> [**Q3**] Currently, TCM/TCMP focus on classification tasks and may not generalize as readily to regression settings, which limits the scope of the paper.
>
> **Ans**: TabCutMix and TabCutMixPlus rely on exchanging features between samples of the same class, which inherently limits their applicability to classification settings. As a result, extending our methods to regression tasks remains a challenge and is beyond the scope of this paper. We have acknowledged this limitation in Limitation Discussion (Appendix. F) of the revised version.
>
> [**Q4**] It would be helpful for the authors to evaluate OOD ratios for Mixup and SMOTE as well, providing a more direct baseline comparison alongside TCM/TCMP (Section D.4.6) Table. 4.
>
> **Ans**: We thank the reviewer for the suggestion. As requested, we have added OOD detection results for Mixup and SMOTE (now included in Table 5). The “Ratio%” column represents the proportion of samples detected as out-of-distribution. While Mixup shows minimal OOD risk, it performs poorly in mitigating memorization compared to our methods. In contrast, TabCutMixPlus achieves a favorable balance—substantially reducing memorization while maintaining low OOD ratios (e.g., 0.36% on Adult vs. 2.06% for TabCutMix and 0.75% for SMOTE), making it a more effective and reliable augmentation strategy for tabular diffusion models.

---

### Official Review · Reviewer_ihym · 2025-03-19

**Overall Recommendation:** 4

**Summary:**

The paper examines memorization in diffusion models for tabular data, providing theoretical insights that demonstrate the optimal score function under the empirical distribution and show that generated data can replicate training samples. To mitigate memorization, the study introduces TabCutMix and TabCutMixPlus, which swap feature segments between training samples to reduce overfitting while preserving data quality. Experimental results confirm that these techniques effectively reduce memorization while maintaining generation quality.

**Claims And Evidence:**

Yes, both the theoretical and empirical results are justified in the work.

**Essential References Not Discussed:**

N/A

**Experimental Designs Or Analyses:**

I think the overall experimental designs make sense and the empirical results are abundant to support the paper's claims.

**Methods And Evaluation Criteria:**

Yes, the authors incorporate a variety of metrics to assess different aspects of the model. For instance, they introduce a memorization score specifically designed for tabular data, along with $\alpha$-precision and $\beta$-recall to evaluate data fidelity. All the chosen metrics are well-justified.

I would suggest including (or at least referencing) important metrics such as $\alpha$-precision and MLE in the main body of the paper rather than leaving them to the appendix.

**Other Comments Or Suggestions:**

N/A

**Other Strengths And Weaknesses:**

**Strengths**
* The experiments are quite comprehensive both in terms of the different scenarios considered and the evaluation metrics used.
* The proposed methods are both intuitive and well discussed.

**Weaknesses**
* Results in Proposition 3.1 is not new, it has been shown in e.g., [1].
* The paper's organization could be improved. Table 1 occupies too much space, pushing important details (for example, the introduction of the evaluation metrics) to the Appendix. And Figure 4 appears after Figure 5, etc.

**Questions For Authors:**

I'm a bit confused about the motivation for using MLE (machine learning efficiency evaluation) as a metric for data fidelity. Can the authors elaborate more here?

**Relation To Broader Scientific Literature:**

Investigating the memorization phenomenon and the approaches of diffusion models is an important topic. While most of other work focus on image data, this work considers tabular data to broaden the scope of such studies, and the proposed techniques are potentially transferrable to other data formats.

**Theoretical Claims:**

I didn’t verify the proof in detail but reviewed the approach used to derive the theoretical results.

---

> ### Author Rebuttal · Authors · 2025-04-01
>
> We thank the reviewer for the constructive comments. All responses and corresponding revisions have been incorporated into the updated manuscript, available at: https://anonymous.4open.science/r/TabCutMix-3F7B/TabCutMix.pdf.
>
> [**Q1**] I would suggest including (or at least referencing) important metrics such as precision and MLE in the main body of the paper rather than leaving them to the appendix.
>
> **Ans**: We have added more descriptions on evaluation metrics and the reference for these important metrics in the main text.
>
> [**Q2**] The Results in Proposition 3.1 are not new, it has been shown in e.g., [1].
>
> **Ans**: We have moved Proposition 3.1 in Appendix A and explicitly cited relevant references to prior works. Our goal, however, is not to claim novelty in the theoretical derivation itself, but rather to contextualize and apply these results specifically to tabular diffusion models.
>
> Our main contribution lies in the empirical investigation of memorization phenomena within tabular diffusion models—a previously unexplored area. Moreover, our proposed novel methods, TabCutMix and TabCutMixPlus, effectively mitigate memorization issues specific to tabular data. We have clarified this distinction explicitly in the revised manuscript to better highlight our empirical findings and methodological contributions.
>
> [**Q3**] The paper's organization could be improved. Table 1 occupies too much space, pushing important details (for example, the introduction of the evaluation metrics) to the Appendix. And Figure 4 appears after Figure 5, etc.
>
> **Ans**: We have reorganized the paper to improve clarity and readability. Specifically, we have moved Magic dataset results from Table 1 to Table 8 (Appendix E.6). We have added evaluation metrics details and reference, in the main text.
>
> [**Q4**] I'm a bit confused about the motivation for using MLE (machine learning efficiency evaluation) as a metric for data fidelity.
>
> **Ans**: MLE evaluates data fidelity by measuring how well a model trained solely on synthetic data performs on real data (e.g., via AUC on a downstream task), which directly reflects the utility of the synthetic data in practical applications. While traditional fidelity metrics like shape score and trend score assess how closely the synthetic data's distribution matches that of the real data, MLE provides an **end-to-end evaluation that incorporates task-specific performance**, offering a complementary perspective on whether the generated data preserves meaningful patterns for model training.

---

### Decision · Program_Chairs · 2025-05-01

**Decision:**

Accept (poster)

**Comment:**

The paper studies memorization in diffusion models for tabular data, showing that the optimal score function under the empirical distribution can lead to regeneration of training samples. To mitigate this, it proposes TabCutMix and TabCutMixPlus, which swap feature segments to reduce overfitting while preserving data quality. Experiments confirm reduced memorization without compromising generation quality.

Most reviewers find the studied problem important, the method can be widely applied, and the evaluation comprehensive.
However, the paper needs a more comprehensive literature review on related works.
Please incorporate the reviewers' feedback during the revision.